# Gut-derived metabolite 3-methylxanthine enhances cisplatin-induced apoptosis via dopamine receptor D1 in a mouse model of ovarian cancer

Zhensheng Mai,[1,2] Yubin Han,[2] Dong Liang,[2] Feihong Mai,[3] Huimin Zheng,[4] Pan Li,[5] Yuan Li,[2] Cong Ma,[2] Yunqing Chen,[2] Weifeng Li,[2] Siyou Zhang,[2] Yinglin Feng,[6] Xia Chen,[4] Yifeng Wang[1]

**ABSTRACT** Platinum-based chemotherapy failure represents a significant challenge in the management of ovarian cancer (OC) and contributes to disease recurrence and poor prognosis. Recent studies have shed light on the involvement of the gut microbiota in modulating anticancer treatments. However, the precise underlying mechanisms, by which gut microbiota regulates the response to platinum-based therapy, remain unclear. Here, we investigated the role of gut microbiota on the anticancer response of cisplatin and its underlying mechanisms. Our results demonstrate a substantial improvement in the anticancer efficacy of cisplatin following antibiotic-induced perturbation of the gut microbiota in OC-bearing mice. 16S rRNA sequencing showed a pronounced alteration in the composition of the gut microbiome in the cecum contents following exposure to cisplatin. Through metabolomic analysis, we identified distinct metabolic profiles in the antibiotic-treated group, with a notable enrichment of the gut-derived metabolite 3-methylxanthine in antibiotic-treated mice. Next, we employed a strategy combining transcriptome analysis and chemical–protein interaction network databases. We identified metabolites that shared structural similarity with 3-methylxanthine, which interacted with genes enriched in cancer-related pathways. It is identified that 3-methylxanthinesignificantly enhances the effectiveness of cisplatin by promoting apoptosis both *in vivo* and *in vitro*. Importantly, through integrative multiomics analyses, we elucidated the mechanistic basis of this enhanced apoptosis, revealing a dopamine receptor D1-dependent pathway mediated by 3-methylxanthine. This study elucidated the mechanism by which gut-derived metabolite 3-methylxanthine mediated cisplatin-induced apoptosis. Our findings highlight the potential translational significance of 3-methylxanthine as a promising adjuvant in conjunction with cisplatin, aiming to improve treatment outcomes for OC patients.

**IMPORTANCE** The precise correlation between the gut microbiota and the anticancer effect of cisplatin in OC remains inadequately understood. Our investigation has revealed that manipulation of the gut microbiota via the administration of antibiotics amplifies the efficacy of cisplatin through the facilitation of apoptosis in OC-bearing mice. Metabolomic analysis has demonstrated that the cecum content from antibiotic-treated mice exhibits an increase in the levels of 3-methylxanthine, which has been shown to potentially enhance the therapeutic effectiveness of cisplatin by an integrated multiomic analysis. This enhancement appears to be attributable to the promotion of cisplatin-induced apoptosis, with 3-methylxanthine potentially exerting its influence via the dopamine receptor D1-dependent pathway. These findings significantly contribute to our comprehension of the impact of the gut microbiota on the anticancer therapy in OC. Notably, the involvement of 3-methylxanthine suggests its prospective utility as

Address correspondence to Yinglin Feng, doctorlynn@126.com, Xia Chen, chenx_fsyyy@163.com, or Yifeng Wang, wangyifeng@smu.edu.cn.

Zhensheng Mai, Yubin Han, and Dong Liang contributed equally to this article. Author order was determined on the basis of seniority.

The authors declare no conflict of interest.

See the funding table on p. 22.

a supplementary component for augmenting treatment outcomes in patients afflicted with ovarian cancer.

**KEYWORDS** 3-methylxanthine, gut microbiota, dopamine receptor D1, cisplatin, ovarian cancer

The American Cancer Society estimated that approximately 19,880 women were newly diagnosed with ovarian cancer (OC), resulting in 12,810 tumor-associated deaths in 2022 (1). OC ranks as the fifth leading cause of cancer-related mortality in women, primarily due to the lack of early disease symptom (1). The management of OC typically involves surgical debulking followed by platinum-based chemotherapy. In cases of disease relapse, patients may be considered for platinum re-treatment. While antiangiogenic treatment with bevacizumab (2) and poly (adenosine diphosphate-ribose) polymerase inhibitors (PARPi) (3) have demonstrated beneficial effects in terms of prolonging progression-free survival and maintenance therapy, respectively, platinum-based regimens remain the most effective chemotherapeutic agents for treating OC. However, the challenges of heterogeneity-associated insensitivity to platinum agents continue to pose significant obstacles, leading to a poor prognosis. Therefore, there is a pressing need to develop innovative and effective drugs that enhance the sensitivity of platinum compounds, while also ensuring tolerability, in order to address this unmet clinical demand.

The gut microbiota has garnered increasing attention due to its influence on the development of anticancer responses. Study have shown that *Bacteroides fragilis* is essential for the immunostimulatory effect of cytotoxic T-lymphocyte antigen 4(CTLA-4) blockade (4). Furthermore, oral supplementation with *Akkermansia muciniphila* after fecal microbiota transplantation (FMT) with programmed death-1 (PD-1) nonresponder feces has been found to restore the efficacy of PD-1 blockade, another form of immunotherapy (5). Beyond immunomodulation, the gut microbiota may also modulate the anticancer responses to traditional chemotherapy agents. For example, *Fusobacterium nucleatum* has been shown to alter the chemotherapeutic response of oxaliplatin in colorectal cancer (6), while gut-derived butyrate has been found to facilitate the efficacy of oxaliplatin by modulating cytotoxic CD8+ T cell immunity (7). Recent studies have also revealed the association between gut microbiota and OC initiation and progression (8, 9). Moreover, OC patients exhibit distinct gut microbiota signatures after platinum-based chemotherapy, and the composition of commensal bacteria differs between platinum-sensitive and platinum-resistant OC patients (10). Disruption of the gut microbiome through antibiotic (ABX) treatment has been shown to result in enhanced tumor growth and altered sensitivity to platinum chemotherapy in preclinical models of OC (11). These findings suggest that the gut microbiota may play a role in modulating the response to anticancer therapy in OC. However, the underlying mechanisms behind these observations require further exploration.

The role of gut-derived metabolites as central regulators in host-microbiota interactions has gained significant attention (12). This highlights the importance of these metabolites in modulating biological functions in distant organs (13–15). Recent studies have confirmed the influence of gut-derived metabolites on gynecological disorders. For instance, patients with polycystic ovary syndrome (PCOS) exhibit elevated levels of *Bacteroides vulgatus* and reduced levels of glycodeoxycholic acid. Supplementing with glycodeoxycholic acid has been shown to improve the PCOS phenotype in mice by inducing IL-22 secretion (16). Furthermore, emerging evidence suggests that gut-derived metabolites can impact cancer development and the response to anticancer therapies. For example, *Lactobacillus gallinarum* produces and catabolizes L-tryptophan, releasing indole-3-lactic acid, which exhibits protective effects against colorectal cancer (17). Similarly, Probio-M9, a strain of *Lacticaseibacillus rhamnosus*, produces beneficial metabolites, including butyrate, which modulate enhanced immunotherapy response

(18). Therefore, identifying key gut-derived metabolites in OC is crucial, as it can provide mechanistic insights into manipulating anticancer therapies.

In this study, we present evidence demonstrating that the depletion of gut microbiota induced by ABXs exacerbates the anticancer efficacy of cisplatin in mice with OC. Metabolomic analysis revealed a significant elevation of 3-methylxanthine in ABX-treated mice. Our findings indicated that metabolites exhibiting structural similarity with 3-methylxanthine interact with genes enriched in cancer-related pathways. Moreover, we identified that 3-methylxanthine enhances cisplatin-induced apoptosis both *in vivo* and *in vitro*. Mechanistically, through comprehensive multiomic analysis, we discovered a positive correlation between 3-methylxanthine and dopamine receptor D1 (DRD1). Subsequently, we demonstrated that both small interfering RNA (siRNA) -targeting DRD1 and pharmacological inhibitors effectively attenuate the augmented cisplatin-induced apoptosis provoked by 3-methylxanthine. These results provide preliminary insights into the mechanism, by which 3-methylxanthine modulates the sensitivity of cisplatin in OC cells. Collectively, our study elucidates the mechanistic role of the gut-derived metabolite 3-methylxanthine in mediating crosstalk between gut microbiota and cisplatin-induced apoptosis. Our data strongly suggest the involvement of gut-derived metabolites in regulating the anticancer efficacy of chemotherapy. Notably, 3-methylxanthine emerges as a safe and promising adjuvant when combined with cisplatin, thereby highlighting its potential translational applications.

## RESULTS

### Gut microbiota dampened the anticancer effect of cisplatin via modulating apoptosis in OC tumor-bearing mice

To investigate the influence of gut microbiota on the anticancer response, we conducted a study to examine whether ABX-induced disruption of gut microbiota affected the efficacy of cisplatin. The experimental design employed in this study is illustrated in Fig. 1A. Female C57BL/6 mice at 6–8 weeks of age were orally administered either phosphate-buffered saline (PBS) or an ABX consisting of ampicillin, neomycin, vancomycin, and metronidazole to deplete commensal bacteria. After 5 days PBS or ABX administration, all mice were intraperitoneally implanted with the murine OC cell line ID8, and tumor formation was monitored using *in vivo* bioluminescence imaging (BLI). ABX treatment was continued every 3 days throughout the experiment in the ABX group. Analysis of cecum content collected at the end point by quantitative polymerase chain reaction (qPCR) targeting the 16S rRNA revealed a significant reduction in the total bacterial load in the ABX-treated group (Fig. 1C). Tumor formation was confirmed by BLI 2 weeks after ID8 cells implantation. There was no significant difference in tumor burden between the ABX-treated and non-treated groups (Fig. 1B and D). Subsequently, all mice intraperitoneally received cisplatin or vehicle treatment. Two weeks after treatment initiation, the ABX-treated mice displayed significant tumor regression in response to cisplatin therapy (Fig. 1B and D), suggesting that gut microbiota may influence therapeutic responses in cancer treatment.

Considering the perturbation of gut microbes by ABX and their impact on the cisplatin response in OC-bearing mice, we proceeded to investigate whether the microbiome composition was altered by cisplatin exposure. We performed 16S rRNA sequencing on cecum content samples from the OC group and the cisplatin group at the endpoint. Principal coordinate analysis (PCoA) revealed distinct separation between the fecal samples of the two groups (Fig. S1A). The composition of gut microbiota exhibited clear alterations in the cisplatin-treated mice compared to the OC group. Specifically, at the phylum level, there was a higher relative abundance of *Proteobacteria* in the cisplatin group (Fig. S1B). Moreover, at the genus level, we observed an increased relative abundance of *Klebsiella* in the cisplatin group compared to the OC group (Fig. S1C). To further identify bacterial taxa that differed significantly between the cisplatin and control groups, we conducted linear discriminant analysis effect size (LEfSe) analysis. *Escherichia* and *Klebsiella* were significantly enriched, while *Paraprevotella* and *Prevotella*

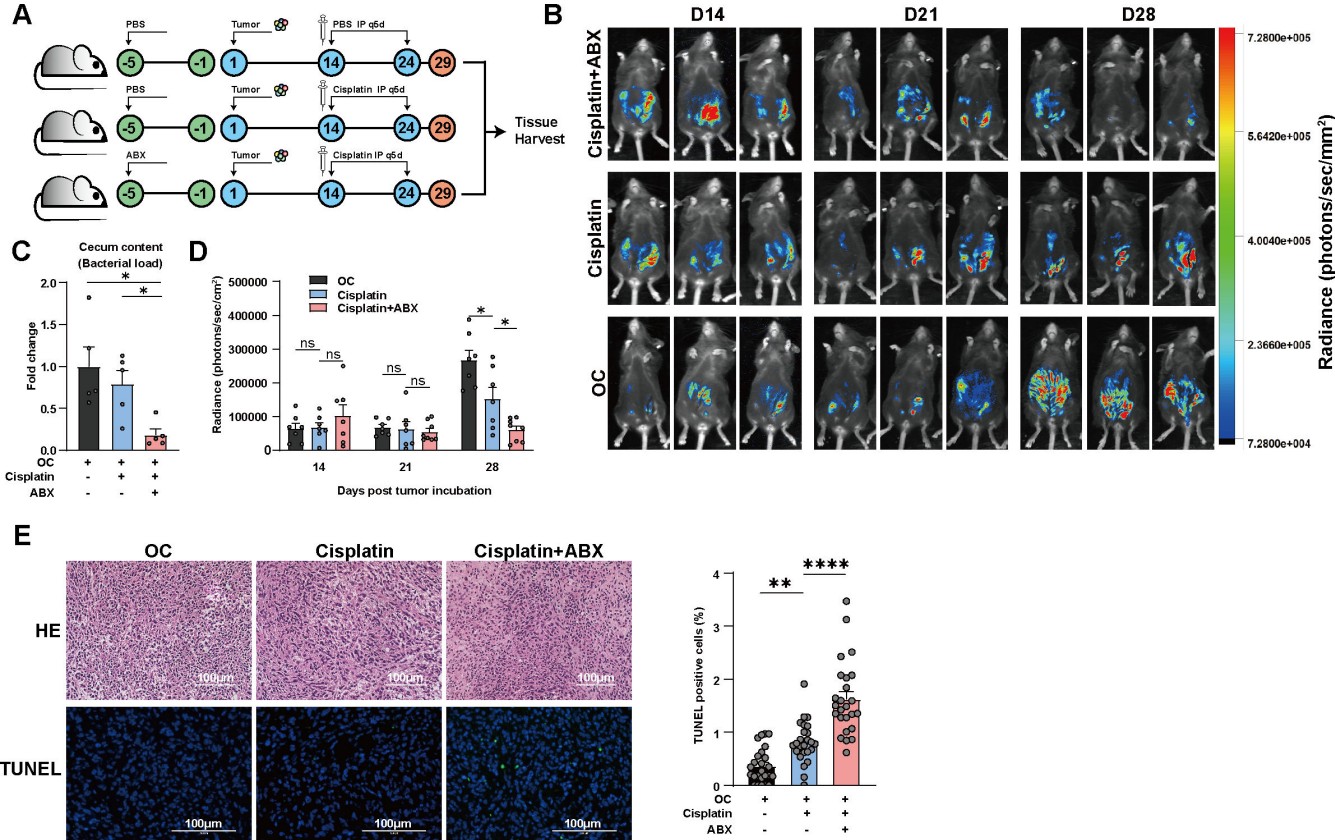

FIG 1 Gut microbiota dampened the anticancer effect of cisplatin via modulating apoptosis in OC tumor-bearing mice. The gut microbiota of the mice was depleted using ABX. Before tumor implantation, the mice were treated with ABXs or PBS for 5 days. ABX treatment was then continued every 3 days after inoculation with ID8 cells. After confirming the presence of tumors through BLI, cisplatin therapy was initiated at a dosage of 5 mg/kg, and administered intraperitoneally every 5 days. The progression of the tumors was monitored by BLI. At the endpoint of the study, the mice were euthanized, and their tumors were surgically removed. Additionally, the content of the cecum was collected. (A) Experimental design. (B) The bioluminescence of mice was observed after the inoculation of luciferase-tagged ID8 cells. The representative images shown depict three mice from each group on days 14, 21, and 28 after tumor incubation. (C) The comparison of the total bacterial load in the cecum contents among the OC, cisplatin, and cisplatin + ABX groups ($n = 5$ per group). (D) Tumor burden was quantified by measuring the total flux. The data for day 14, 21, and 28 after tumor incubation are presented ($n = 7$–8 per group). (E) The upper panel shows a representative image of H&E staining of tumor tissue. The lower panel displays the TUNEL staining of the tumor tissue, with TUNEL-positive cells shown in green, and TUNEL-negative cells stained with DAPI. For quantification, TUNEL-positive and TUNEL-negative cells were counted in five random fields ($n = 5$ per group). Scale bar = 100 µm. Error bars represent SEM. $*P < 0.05$; $**P < 0.01$; $****P < 0.0001$, ns, no significant difference.

were depleted in the cisplatin-treated mice (Fig. S1D and E). These results demonstrate that cisplatin treatment alters the composition of gut microbiota in OC-bearing mice.

Cisplatin, an anticancer agent, exerts its effects by binding to DNA and forming intra- and inter-stranded crosslinks, leading to DNA damage and subsequent induction of apoptosis (19). To assess whether ABX treatment enhanced cisplatin-induced apoptosis, we performed Terminal deoxynucleotidyl transferase dUTP nick end labeling (TUNEL) staining on ID8 tumor samples harvested from tumor-bearing mice at the endpoint. Histological analysis of hematoxylin and eosin (H&E) staining was also conducted to confirm the OC tumor phenotype (Fig. 1E, upper panel). The percentage of apoptotic cells in the cisplatin + ABX group was significantly higher compared to the cisplatin group, indicating that depletion of gut microbes by ABXs facilitates cisplatin-induced apoptosis in OC cells (Fig. 1E, lower panel). Collectively, our findings suggest that gut microbiota dampen the anticancer effect of cisplatin through modulation of apoptosis.

## Gut-derived metabolite was altered by ABX treatment

Given the distinct separation observed in gut microbiota under cisplatin exposure, it is crucial to consider the role of gut-derived metabolites as important intermediaries in the gut-host crosstalk (12). Thus, our objective was to investigate the metabolomic changes in the cecum content of tumor-bearing mice treated with cisplatin + ABX or cisplatin alone at the endpoint. The administration of ABXs not only had a profound impact on the gut microbiota but also induced significant alterations in the gut metabolome. We identified changes in metabolite levels ranging from twofold to over 71707-fold (Table S1). Orthogonal partial least squares discriminant analysis (OPLS-DA) displayed a distinct distribution pattern in the cisplatin + ABX group compared to the cisplatin group, as illustrated in Fig. 2A. Furthermore, the volcano plot depicted the enrichment of differentially expressed metabolites between the two groups (Fig. 2B). Notably, several metabolites with potential roles in cell apoptosis, such as thioguanine, genistein, 3-methylxanthine, (−)-epiafzelechin, and quinic acid showed marked elevation in the cisplatin + ABX group based on the log2 fold change (Fig. 2C) (20–23).

In summary, our findings indicate that ABX treatment induced substantial metabolomic changes in the cecal content of tumor-bearing mice. These alterations highlight the intricate interplay between gut microbiota and host metabolism, potentially contributing to the observed effects on tumor response and apoptosis induction.

## The level of 3-methylxanthine was elevated in ABX-treated mice

We proceeded to investigate the potential involvement of gut-derived metabolites in the regulation of the anticancer effect. The metabolic pathway enrichment analysis of differential metabolites revealed a notable enrichment within the caffeine metabolism pathway (Fig. 3A). Notably, our investigation identified a substantial enrichment of 3-methylxanthine, a constituent of the caffeine metabolism pathway, within the cisplatin + ABX group. Additionally, an alteration in the levels of other metabolites within the caffeine metabolism pathway was also observed. Specifically, there was a notable increase in the levels of caffeine, 3.7-dimethyluric acid, and 3-methylxanthine. Conversely, downstream metabolites in this pathway, such as xanthine and xanthinosine, showed a significant decrease (Fig. 3B). This finding suggests that ABX treatment may disrupt the caffeine metabolism pathway, subsequently leading to the accumulation of 3-methylxanthine in the body.

Methylxanthine derivatives, including 3-methylxanthine, have been identified as adenosine receptor antagonists and have demonstrated molecular functions related to phosphodiesterase inhibition (24). Applying a strategy outlined in Fig. 3C, we aimed to explore the underlying association between 3-methylxanthine and the anticancer efficacy of cisplatin.

In the first arm of our investigation, we hypothesized that tumorous functional and phenotypical alterations would coincide with changes in gut-derived metabolites. To explore functional changes in tumor tissues of matched mice, we performed transcriptome sequencing. Principal component analysis (PCA) revealed significant differences in gene profiles between the cisplatin + ABX and cisplatin groups (Fig. S2A). We observed an increased expression of 377 genes and a decreased expression of 103 genes under ABX treatment (Fig. S2B). To further analyze these gene expression patterns, we conducted gene set enrichment analysis (GSEA) (25). Interestingly, GSEA indicated a significant enrichment of the cyclic adenosine monophosphate (cAMP) signaling pathway in cisplatin + ABX-treated mice (GSEA normalized enrichment score = 1.89, $P = 0.002$) (Fig. 3D). Given that 3-methylxanthine possesses a molecular function of phosphodiesterase inhibition and can increase intracellular cAMP concentrations, which are involved in the cAMP signaling pathway, these findings confirmed our assumption regarding the pivotal role of 3-methylxanthine in modulating the response to cisplatin.

In the second arm of our investigation, we postulated that metabolites sharing structural similarity might be involved in similar biological functions (26, 27). Employing PubChem, we searched for structurally similar metabolites to 3-methylxanthine and

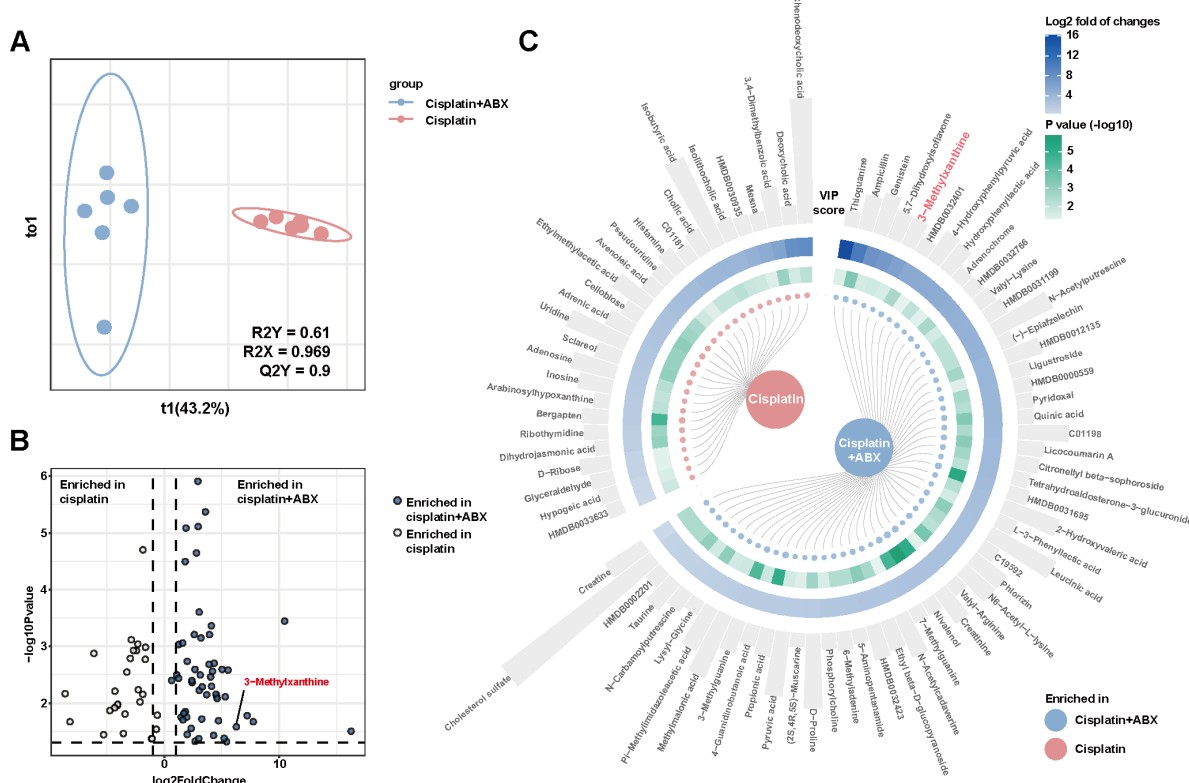

**FIG 2** Gut-derived metabolite was altered by ABX treatment. The cecum content of tumor-bearing mice at the endpoint was analyzed using LC-MS to investigate the metabolomic changes in both the cisplatin + ABX and cisplatin groups. (A) OPLS-DA was performed to analyze the metabolomics of the cecum content. (B) Volcano plot analysis revealed that the level of 3-methylxanthine was significantly increased in cisplatin + ABX-treated mice. (C) The results indicated differential metabolites between the cisplatin + ABX and cisplatin groups, and they were arranged based on log2 fold change. The height of bars indicated VIP scores of metabolites.

identified the top 10 relevant metabolites that interacted with 20 genes enriched in apoptosis and caffeine metabolism (Fig. 3E; Table S2–S4).

In order to explore the relationship between gut microbiota and 3-methylxanthine, we evaluated the level of 3-methylxanthine in feces of germ-free mice and specific pathogen free (SPF) mice by high performance liquid chromatography (HPLC) (Fig. 3F). Compared with SPF mice, the level of 3-methylxanthine was significantly higher in germ-free mice (Fig. 3G). Next, we explored the change of level of 3-methylxanthine in the feces of SPF mice after ABX treatment and FMT. Subsequent to the ABX treatment, SPF mice exhibited an increased concentration of 3-methylxanthine (Fig. 3H), corroborating our metabolomic analyses that suggested a disruption in the microbial metabolism of 3-methylxanthine. Notably, a reduction in 3-methylxanthine levels was discerned following FMT (Fig. 3H), further endorsing the contributory role of gut microbiota in this metabolic pathway. The results of qPCR demonstrated a significant alteration in bacterial load in mice following administration of ABX and FMT (Fig. 3I). In summary, our finding suggests that depletion of gut microbiota may lead to an elevation in the levels of 3-methylxanthine.

## 3-Methylxanthine improved therapeutic efficacy of cisplatin in both immunocompetent and immunocompromised OC-bearing mice

Given the established potent anticancer effect of 3-methylxanthine, our hypothesis centered on the potential therapeutic benefits of 3-methylxanthine treatment in conjunction with cisplatin therapy (28). To assess the role of 3-methylxanthine, we utilized an ID8-bearing C57BL/6 mouse model. Tumor formation was confirmed through

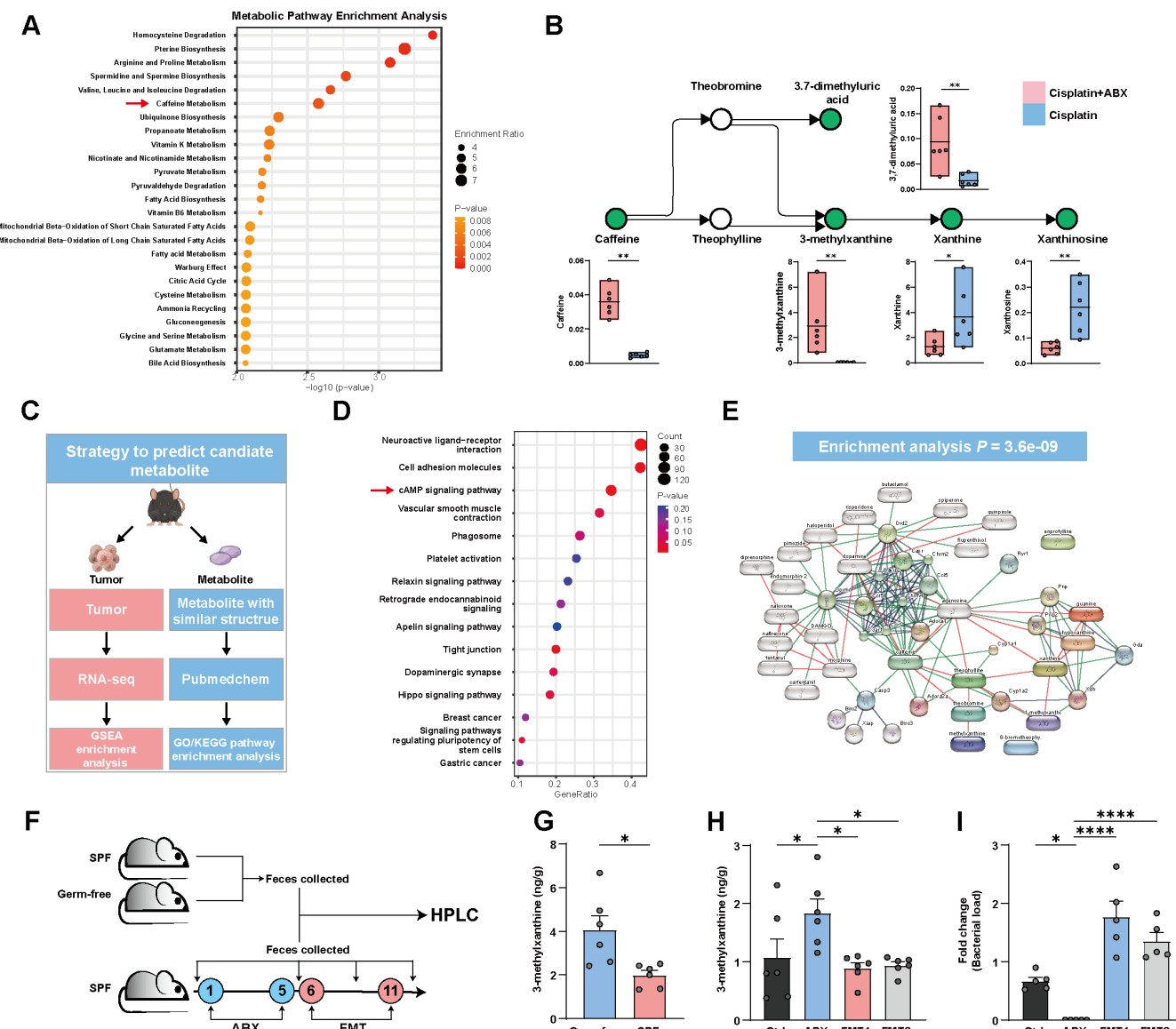

**FIG 3** The level of 3-methylxanthine was elevated in ABX-treated mice. (A) Metabolic pathway enrichment analysis based on differential metabolites. (B) Partial view of the caffeine metabolism pathway highlighting metabolite abundances detected in metabolomics data. (C) Flow chart to verify the function of 3-methylxanthine. (D) GSEA showed a significance enrichment in cAMP signaling pathway in cisplatin + ABX group. (E) Metabolites that share structural similarity with 3-methylxanthine, and their respective interaction genes as mapped to those in the STITCH database. (F) Experimental design. (G) The concentration of 3-methylxanthine in the feces of germ-free and SPF mice (*n* = 6 per group). (H) Feces of SPF mice were collected after ABX treatment and FMT. The level of 3-methylxanthine in the feces was measured by HPLC (*n* = 6 per group). (I) Total fecal bacterial load (*n* = 5 per group). ABX, feces collected after ABX treatment; ctrl, feces collected before ABX treatment; FMT, feces collected 3 days after FMT; FMT2, feces collected 6 days after FMT. Error bars represent SEM. *P < 0.05; **P < 0.01, ****P < 0.0001.

BLI 2 weeks after intraperitoneal implantation of ID8 cells. Subsequently, the mice were randomly assigned to receive either PBS, 3-methylxanthine, cisplatin or cisplatin + 3-methylxanthine treatment (Fig. 4A). After 2 weeks of treatment initiation, tumor burden was assessed using BLI. The results indicated a significant reduction in tumor progression in mice treated with cisplatin + 3-methylxanthine compared to cisplatin alone 2 weeks after treatment initiation. Conversely, no significant tumor regression was observed in mice treated with 3-methylxanthine alone or the vehicle control (Fig. 4B and C).

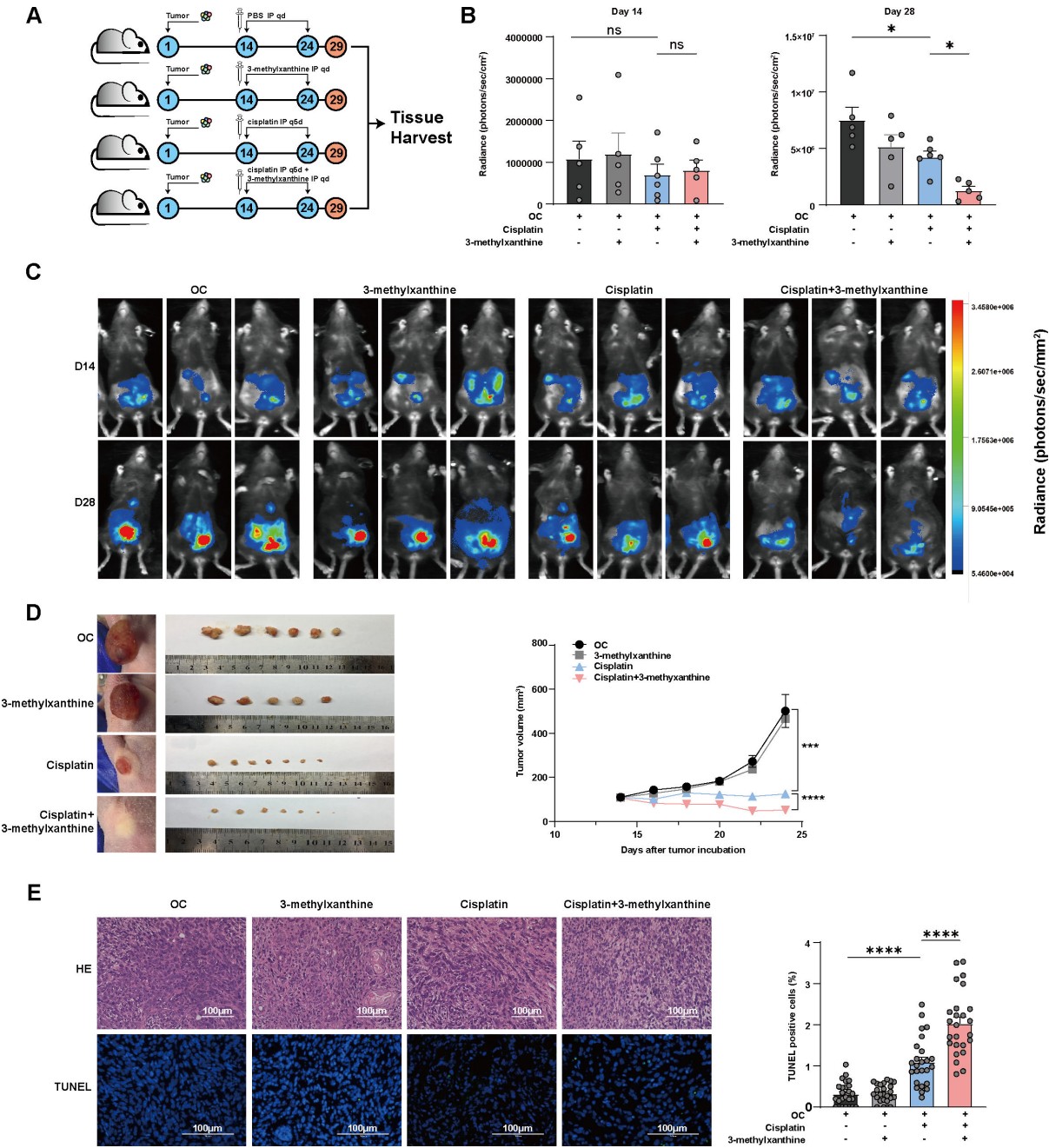

**FIG 4** 3-Methylxanthine improved anticancer efficacy of cisplatin in both immunocompetent and immunocompromised OC-bearing mice. C57BL/6 mice were injected intraperitoneally with ID8 cells to establish immunocompetent tumor model. BALB/c-nu mice were injected subcutaneously with ID8 cells to establish immunocompromised tumor model. 14 days after tumor implantation, tumor formation was confirmed in both models. Mice were randomized and then treated with (i) PBS, (ii) 3-methylxanthine (10 mg/kg, intraperitoneally, daily), (iii) cisplatin (5 mg/kg, intraperitoneally, every 5 days), or (iv) cisplatin + 3-methylxanthine. Tumor growth was monitored by BLI in intraperitoneal xenograft model and by measuring tumor volume in subcutaneous xenograft model. At the end point, all mice were euthanized, and their tumors were excised. (A) Experimental design. (B) Tumor burden was quantified by measuring the total flux. Data from day 14 and 28 after tumor implantation are presented ($n$ = 5–6 per group). (C) BLI of tumor burden was performed in the C57BL/6 immunocompetent tumor model on day 14 and 28 after tumor implantation. The presented images are representative of three mice in each group. (D) ID8 cells were subcutaneously inoculated in the right flanks of BALB/c-nu immunocompromised mice. Tumor growth curves were constructed based on tumor volumes measured every 2 days, starting from 14 days after tumor implantation. (E) A representative image of H&E staining of tumor tissue is shown in the upper panel. TUNEL staining of the tumor tissue is shown in the lower panel. TUNEL-positive cells were labeled in green, and TUNEL-negative cells were stained with DAPI. The counts were conducted in five random fields ($n$ = 5 per group). Scale bar = 100 µm. Error bars represent SEM. *$P$ < 0.05; ***$P$ < 0.001; ****$P$ < 0.0001; ns, no significant difference.

Recent studies have emphasized the immunomodulatory role of the gut microbiota in the context of antitumor responses (4, 5). Gut-derived metabolites, including short-chain fatty acids, have also been implicated in influencing autoimmune responses (29). Considering these findings, we aimed to explore whether the anticancer effect of 3-methylxanthine depended on an intact immune system. To investigate this, we employed Bagg albino/c nude (BALB/c-nu) immunocompromised mice. Tumor formation was induced by subcutaneously injecting ID8 cells into BALB/c-nu mice. Tumor confirmation occurred 2 weeks after tumor implantation and then treatment initiation. Our observations revealed a significant tumor regression, as measured by tumor volume, in the cisplatin + 3-methylxanthine group compared to the cisplatin group (Fig. 4D). This suggests that the anticancer effect mediated by 3-methylxanthine is independent of the innate immune system. H&E stained tissue sections were used to confirm the OC tumor phenotype (Fig. 4E, upper panel). Furthermore, we noted a significant increase in TUNEL-positive cells in the cisplatin + 3-methylxanthine group compared to the cisplatin group, indicating that 3-methylxanthine-induced tumor regression relies on the induction of apoptosis (Fig. 4E, lower panel).

To assess the safety of 3-methylxanthine treatment, we evaluated its potential toxicological effects on major organs, such as the liver and kidneys, by measuring serum creatinine (Crea), alanine aminotransferase (ALT), and aspartate transaminase (AST) levels. Our data indicated that 3-methylxanthine treatment was well tolerated without any discernible toxicological effects on these organs (Fig. S3A). This further underscores its potential implications for clinical treatment. In summary, our findings demonstrate that 3-methylxanthine promotes cisplatin-induced apoptosis in both immunocompetent and immunocompromised OC-bearing mice.

## 3-Methylxanthine promoted cisplatin-induced apoptosis in OC cell lines

Given the observed synergistic effect of 3-methylxanthine with cisplatin *in vivo*, which was found to be independent of the immune system, we postulated that 3-methylxanthine might directly exert an antitumor effect on cancer cells. To investigate this, we proceeded to evaluate the effect of 3-methylxanthine in combination with cisplatin. A cell viability assay was performed in the presence or absence of 3-methylxanthine, revealing that 3-methylxanthine sensitized ID8 and SKOV3 cells to the growth inhibitory effects of cisplatin. The IC50 values of cisplatin in ID8 and SKOV3 cells were determined to be 21.21 and 37.5 µM, respectively, after 24 h of treatment. Remarkably, in combination with 3-methylxanthine, the IC50 values of cisplatin decreased to 12.72 and 25.42 µM, respectively (Fig. 5A). Furthermore, EdU proliferation assays demonstrated that the combination of 3-methylxanthine with cisplatin synergistically inhibited cell proliferation in both cell lines *in vitro* (Fig. 5B). These results strongly suggest that 3-methylxanthine enhanced the inhibitory effect of cisplatin on proliferation in both murine and human OC cells.

Subsequently, flow cytometry was performed to examine apoptosis, revealing that 3-methylxanthine alone did not induce apoptosis *in vitro*. However, the combination of 3-methylxanthine with cisplatin synergistically induced apoptosis *in vitro* (Fig. 5C). To further investigate the underlying mechanisms, we assessed the expression of key proteins using western blot analysis. γH2AX, a well-known marker of DNA damage, was analyzed, and the results demonstrated that combination therapy with 3-methylxanthine and cisplatin significantly increased γH2AX protein expression compared to cisplatin alone. Furthermore, the induction of cisplatin-induced apoptosis by 3-methylxanthine was supported by increased protein expression of the cleaved (active) forms of caspase-3. Additionally, the expression levels of the apoptosis-related proteins Bcl-2 and Bax were evaluated. The results indicated that 3-methylxanthine decreased the expression of Bcl-2, while the expression of Bax remained unchanged (Fig. 5D). Collectively, these findings indicate that 3-methylxanthine synergistically enhances cisplatin-induced apoptosis in both murine and human OC cells.

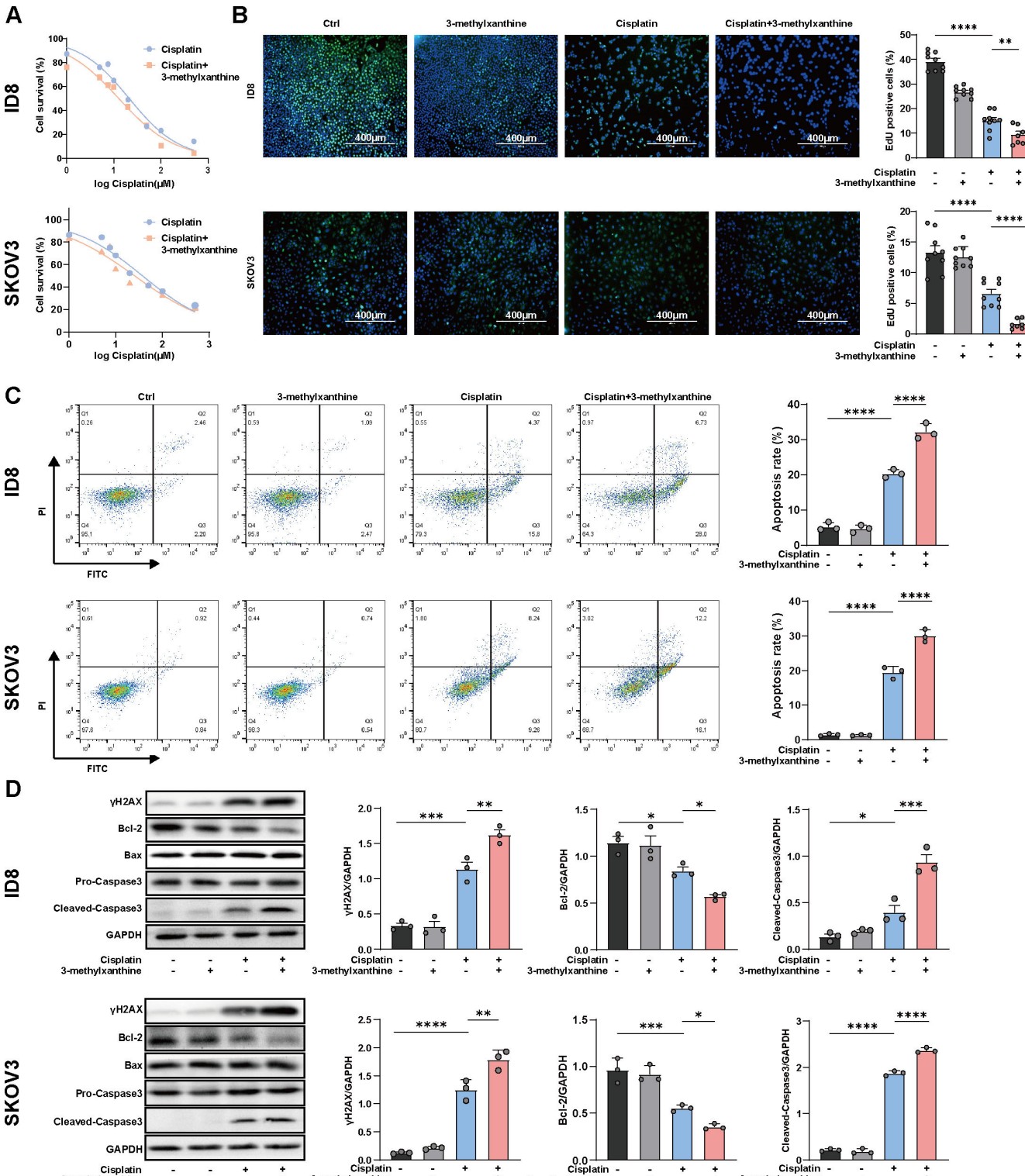

**FIG 5** 3-Methylxanthine promoted cisplatin-induced apoptosis in OC cell lines. (A) 3-Methylxanthine (2 mM) co-treatment sensitized ID8 and SKOV3 cells to the growth inhibitory effect of cisplatin. (B) The EdU assay was performed on ID8 and SKOV3 cells after incubation with 3-methylxanthine (2 mM) and/or cisplatin (20 µM). EdU-positive cells were labeled in green, and EdU-negative cells were stained with DAPI. The counts were conducted in three random fields, with three replicates per group. Scale bar was 400 µm. (C) Apoptosis assay ($n = 3$ per group). (D) Western blot analysis ($n = 3$ per group). Error bars represent SEM. *$P < 0.05$; **$P < 0.01$; ***$P < 0.001$; ****$P < 0.0001$.

# 3-Methylxanthine enhanced cisplatin-induced cancer apoptosis via DRD1

Having identified 3-methylxanthine's involvement in the anticancer effect of cisplatin, we aimed to investigate the underlying mechanism. 3-Methylxanthine, a derivative of methylxanthine, is known to exert its molecular effects through adenosine receptor antagonism and phosphodiesterase inhibition (30). Inhibition of phosphodiesterase leads to increased intracellular cAMP concentrations. Consequently, we focused our attention on genes involved in the cAMP signaling pathway. A heatmap was generated to illustrate the relative expression of genes associated with the cAMP signaling pathway, including *DRD1, Lipe, Adcy10, Sucnr1, Cnga4*, and *Hhip*. These genes exhibited upregulation in the cisplatin + ABX-treated mice compared to the cisplatin group (Fig. S4A). By employing an integrated multiomics analysis approach, we were able to validate the association between the abundance of metabolites implicated in caffeine metabolism and the expression levels of genes related to the cAMP signaling pathway in matched

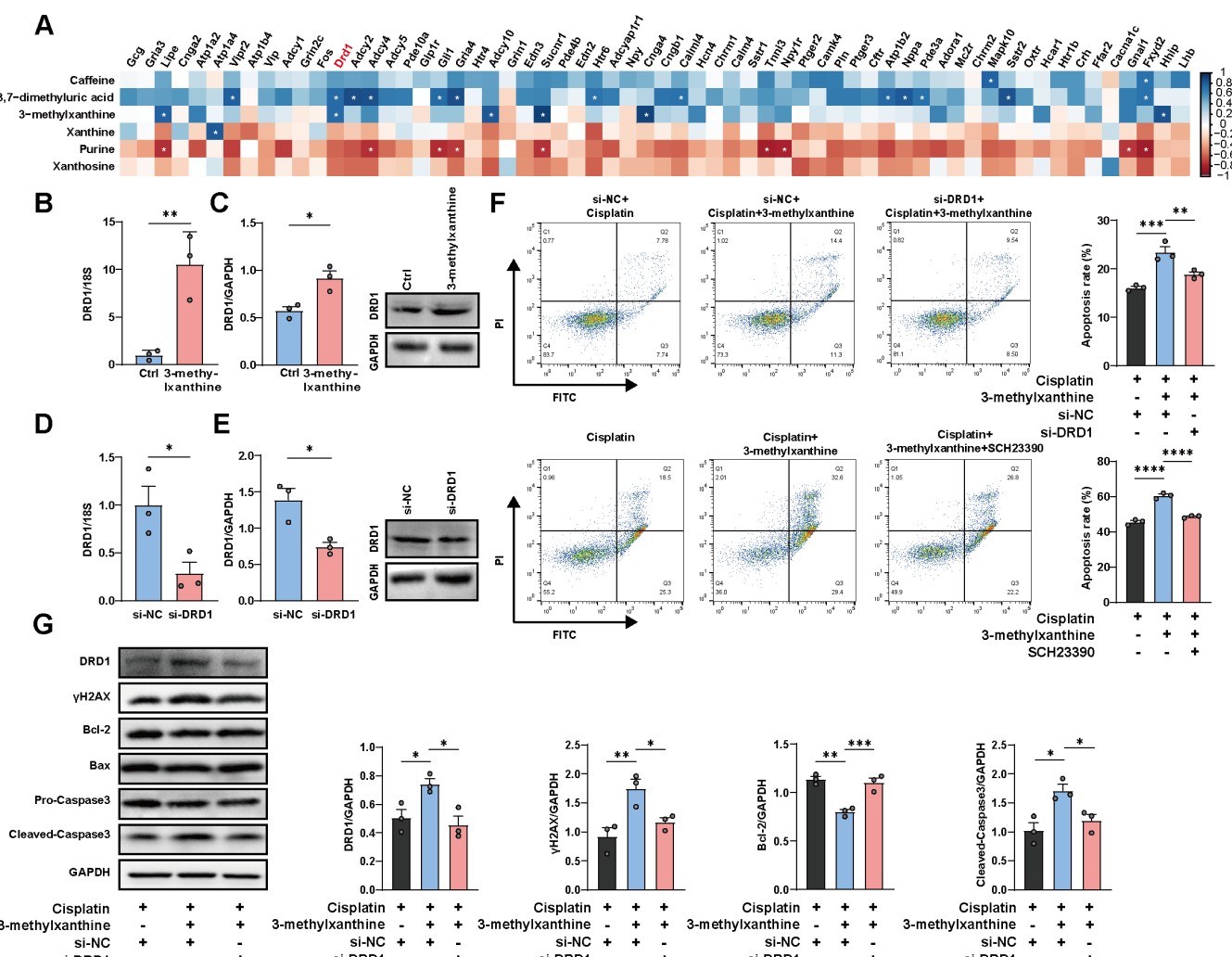

**FIG 6** 3-Methylxanthine enhanced cisplatin-induced cancer apoptosis via DRD1. (A) Correlation analysis revealed a positive correlation between the abundance level of 3-methylxanthine and cAMP signaling pathway related genes. (B and C) ID8 cells were treated with 3-methylxanthine (2 mM) for 24 h. qRT-PCR and Western blotting showed the expression of DRD1 (*n* = 3 per group). (D and E) ID8 cells were transfected with siRNA of DRD1 or si-NC. qRT-PCR and western blotting were conducted to measure the expression of DRD1 (*n* = 3 per group). (F) ID8 cells were transfected with siRNA of DRD1 or si-NC and then treated with 3-methylxanthine and/or cisplatin for 24 h (upper panel). In another experiment, ID8 cells were treated with the DRD1 antagonist (SCH23390) and 3-methylxanthine and/or cisplatin for 24 h. After treatment, cells were harvested and subjected to apoptosis assay (*n* = 3 per group). (G) ID8 cells were transfected with siRNA of DRD1 or si-NC and then treated with 3-methylxanthine and/or cisplatin for 24 h. The cells were harvested and were subjected to western blotting assay with the indicated antibodies (*n* = 3 per group). Error bars represent SEM. *$P < 0.05$; **$P < 0.01$; ***$P < 0.001$.

samples, as depicted in Fig. 6A. Notably, we observed a significant positive correlation between 3-methylxanthine and DRD1 (Fig. S5A). *In vitro* studies using quantitative reverse transcription polymerase chain reaction (qRT-PCR) and western blotting validated that 3-methylxanthine upregulated DRD1 expression (Fig. 6B and C).

DRD1 is a member of the dopamine receptor family, has been implicated in cancer progression (31). We hypothesized that DRD1 could serve as the target of 3-methylxanthine. To further confirm the involvement of DRD1 in mediating the effects of 3-methylxanthine, we conducted loss-of-function studies in ID8 cells using both genetic and pharmacological approaches. Apoptosis was assessed by flow cytometry and western blotting following transfection with either non-coding siRNA (nc-siRNA) or siRNA-targeting DRD1. We observed a significant reduction in DRD1 mRNA and protein levels following siRNA-targeting DRD1, whereas no reduction was observed with nc-siRNA (Fig. 6D and E). Importantly, siRNA against DRD1 demonstrated a significant decrease in apoptosis in ID8 cells subjected to co-treatment with 3-methylxanthine and cisplatin, while nc-siRNA had no effect (Fig. 6F, upper panel). Consistent with the apoptosis assay, siRNA against DRD1 significantly reduced γH2AX and cleaved caspase-3 expression, and increased the expression of Bcl-2 in ID8 cells following co-treatment with 3-methylxanthine and cisplatin (Fig. 6G). Complementary studies were conducted using the pharmacological inhibitor SCH23390, which targets DRD1. Incubation of ID8 cells with SCH23390 substantially attenuated the apoptosis induced after co-treatment with 3-methylxanthine and cisplatin. (Fig. 6F, lower panel). Thus, both genetic and pharmacological loss-of-function studies confirm that 3-methylxanthine enhances cisplatin-induced apoptosis through DRD1.

## DISCUSSION

OC patients initially exhibit a favorable response to surgical debulking and platinum-based chemotherapy. However, relapse with platinum-resistant cancer frequently occurs, and the effectiveness of novel immunotherapy in OC is limited, with a response rate below 10% (32). Given these challenges, there is a critical need to develop innovative, effective, and well-tolerated drugs that can enhance the chemosensitivity of platinum compounds. The potential role of gut microbiota in regulating the anticancer efficacy of platinum in OC remains poorly understood. Addressing this knowledge gap, our study aimed to investigate the impact of gut microbiota perturbation induced by ABXs on the anticancer efficacy of cisplatin in mice bearing OC. Through comprehensive multiomic studies, we have identified a key gut-derived metabolite, 3-methylxanthine, as a significant regulator in modulating the response to cisplatin. Notably, our further investigations have confirmed that 3-methylxanthine enhanced cisplatin-induced apoptosis through the DRD1 pathway. These findings underscore the potential significance of a "gut–ovary" axis in regulating the effectiveness of anticancer therapies, with gut-derived metabolites demonstrating promise as safe and promising small molecule compounds for enhancing the sensitivity of cisplatin. By elucidating the complex interplay between gut microbiota, metabolites, and platinum chemotherapy, our study contributes to a deeper understanding of the underlying mechanisms and offers novel insights into therapeutic strategies that can improve the treatment outcomes for OC patients.

Cisplatin, a chemotherapy agent extensively employed in the treatment of OC, has been shown to exert deleterious effects on the gut microbiota (33). The present investigation corroborates the notion that cisplatin administration induces substantial alterations in the microbial composition of mice afflicted with OC, characterized notably by an augmented abundance of *Escherichia, Klebsiella,* and *Proteus*. It is noteworthy that these microorganisms predominantly belong to the gram-negative bacterial category. Notably, perturbations in the gut microbiota have also been observed in patients with OC and platinum resistance (10).

The implications of cisplatin-induced dysbiosis are currently under investigation. Nevertheless, emerging evidence suggests that alterations in the gut microbiota may

have the potential to influence both the effectiveness and adverse effects of platinum-based treatments. One proposed mechanism is the modulation of systemic oxidative stress through the interaction between gut microbiota and tumor-infiltrating myeloid-derived cells, affecting reactive oxygen production and subsequently impacting platinum sensitivity (34). Additionally, gut microbial metabolites, such as butyrate, have been shown to enhance the therapeutic efficacy of oxaliplatin by regulating CD8+ T cell immunity. However, the immunomodulatory effect of butyrate treatment does not seem to affect the sensitivity of cisplatin (7). The precise mechanisms, by which gut microbiota regulate the efficacy of cisplatin in OC, remain unclear. Therefore, further mechanistic insights are necessary to understand the relationship between gut microbiota and platinum response. In light of this, our hypothesis posits that dysbiosis of the gut microbiota may contribute to alterations in cisplatin response. To explore this, our study employed broad-spectrum ABXs to deplete the gut microbiota, and it was confirmed that ABX treatment significantly enhanced the anticancer efficacy of cisplatin in mice with OC. This phenomenon was also observed with the chemotherapeutic drug gemcitabine and doxorubicin, as ABX treatment-induced gut microbiota depletion was associated with an improved response rate in a preclinical model (35, 36). Gut-derived metabolites serve as key regulators between the gut microbiota and the host, influencing various aspects of host physiology (12). Metabolomic analysis revealed distinguishable metabolites derived from the intestinal microbiota in mice stimulated with cisplatin + ABX compared to the cisplatin group. Interestingly, 3-methylxanthine exhibited a prominent increase in the cecum content of mice treated with cisplatin + ABX. Our focus on 3-methylxanthine was based on multiple considerations, with a primary factor being its significant upregulation in the cisplatin + ABX treatment group. 3-Methylxanthine is an important metabolite in the caffeine metabolism pathway and is among the differential metabolites enriched in this pathway. Additionally, through the functional prediction of metabolites, 3-methylxanthine and its structurally similar metabolites had been found to interact with genes related to apoptosis, further supporting the emphasis on 3-methylxanthine. *In vivo* experiments confirmed that supplementation with 3-methylxanthine significantly enhanced the response to cisplatin in mice. However, it is worth noting that these metabolites, such as thioguanine and genistein, have been reported to have apoptotic-inducing effects (20, 22). The potential contribution of these metabolites to the anticancer effects of cisplatin cannot be completely ruled out.

Recent studies have focused on the modulation of the immune system's response to chemotherapy by the gut microbiota. The effectiveness of 3-methylxanthine was also observed in immunocompetent mice, suggesting a direct interaction between 3-methylxanthine and cisplatin, altering the anticancer response of cisplatin. In contrast to our findings, Chambers et al. identified perturbation of the gut microbiota through ABX treatment, which promoted tumor growth and suppressed cisplatin sensitivity (11). We believe that the differences in experimental results are primarily caused by different ABX treatment plans. Previous studies have demonstrated that various ABX schemes have differing effects on the eradication of gut microbiota, which in turn have shown different outcomes in the progression of OC (9, 11). Additionally, these variances could also stem from methodological differences in tumor status assessment, differences in intervention measures, and the timing of cisplatin treatment schemes. More research is necessary to confirm microbiota and metabolomic changes in OC patients responding differently to platinum-based therapies. By identifying key bacterial strains and metabolites, we can validate our findings and better understand the role of gut microbiota in disease development and treatment.

Metabolomic profiling revealed that the differential metabolites enriched in the caffeine metabolism pathway and the level of metabolites involving in caffeine metabolism were altered in ABX-treated mice, suggesting that ABX treatment could induce dysbiosis of gut microbiota that might disrupt caffeine metabolism. Methylxanthine is primarily metabolized in the liver by enzymes known as cytochrome P450 (CYP) enzymes, specifically CYP1A2 (37). Emerging research suggests that gut microbiota can

influence caffeine metabolism primarily in the intestinal lumen before its absorption into the bloodstream. Gut bacteria utilized N-demethylase breakdown caffeine into related methylxanthine (38). ABX treatment can also attenuated caffeine degradation in *Hypothenemus hampei* (39). Our research verified a notable rise in 3-methylxanthine levels in germ-free mice. Additionally, the levels of 3-methylxanthine in the feces of SPF mice were altered after treatment with ABX and FMT. These findings provide further evidence supporting our hypothesis that gut microbiota modulation can influence caffeine metabolism, potentially affecting 3-methylxanthine levels. Through an in-depth metabolomics analysis, we observed a significant decrease in the concentrations of metabolites related to purine metabolism, such as xanthine, xanthinosine, inosine, cAMP, and cGMP, in mice with ABX-induced gut microbiota depletion. This trend aligned with observations in germ-free mice, further supporting the conclusion that gut microbiota may intervene in purine metabolism (40). It is noteworthy that both caffeine metabolism and purine metabolism share the intermediate metabolite xanthine, suggesting a possible convergence in metabolic pathways. Further research is warranted to elucidate the intricate relationship between gut microbiota and the metabolism of caffeine and purines, enhancing our overall understanding in this area.

Methylxanthines, including caffeine, theophylline, and theobromine, are commonly found in various beverages and foods in our daily diet (37). These compounds have been extensively utilized as therapeutic agents and widely employed as central nervous system stimulants, bronchodilators, coronary dilators, and adjuvant treatments in anticancer therapies (41). Methylxanthines have been shown to affect multiple molecular targets involved in DNA repair and cell cycle regulation (42, 43), potentially influencing the response of cancer cells to cisplatin. Of particular interest in our investigation is 3-methylxanthine, an inhibitor of cyclic nucleotide phosphodiesterase, an enzyme responsible for degrading cAMP (30). Our focus on 3-methylxanthine stemmed from our transcriptomics analysis, which revealed a significant enrichment in the cAMP signaling pathway. Furthermore, chemical-protein interaction enrichment analysis confirmed that metabolites sharing structural similarity with 3-methylxanthine interacted with genes enriched in cancer-related pathways. These findings suggested that 3-methylxanthine plays a major role in enhancing apoptosis. In our study, we present the first evidence that 3-methylxanthine can induce apoptosis in combination with cisplatin through a Bcl-2-dependent pathway. Our results align with previous studies demonstrating synergistic effects of other methylxanthines, such as caffeine and theophylline with cisplatin (44, 45). These findings provide valuable preclinical evidence regarding the translational potential of 3-methylxanthine. However, the optimal dosage, timing, and combination strategies involving 3-methylxanthine and platinum-based chemotherapy drugs remain to be determined. We believe that future studies focusing on the supplementation of 3-methylxanthine derivatives will enhance the efficacy of anticancer therapies.

Our study has undertaken a comprehensive analysis of the underlying mechanisms, through which 3-methylxanthine induces apoptosis in a murine OC cell line. Utilizing both bioinformatic and functional approaches, we have successfully elucidated that the modulation of apoptosis in OC cells by 3-methylxanthine is mediated via the DRD1. This conclusion is supported by the findings that both specific knockdown of DRD1 using siRNA and the administration of a DRD1 antagonist effectively blocked the apoptotic phenotypes induced by 3-methylxanthine. Notably, 3-methylxanthine functions as an inhibitor of cyclic nucleotide phosphodiesterase, an enzyme responsible for cAMP degradation, as well as an antagonist of adenosine receptors (30). Our investigation highlights the intricate interplay between adenosine and dopamine receptors. Adenosine receptors are colocalized with dopamine receptor and functionally antagonizes dopamine signaling (46, 47). The potentiation of dopamine receptor effects by adenosine antagonists can be attributed, at least in part, to intramembrane interactions between specific subtypes of dopamine and adenosine receptors (48). Similar phenomena have been observed in other methylxanthines, such as caffeine, which blocks

adenosine A1 receptors, resulting in enhanced DRD1 signaling (49). Notably, DRD1 signaling has been implicated in inflammation and apoptosis via the cAMP signaling pathway (50, 51). Moreover, the cAMP signaling pathway has been shown to significantly impact the enhancement of cisplatin sensitivity and reversal of cisplatin resistance in OC cells (52, 53). In conclusion, our study provides compelling evidence regarding the role of 3-methylxanthine in mediating apoptosis in a murine OC cell line. Specifically, we have demonstrated the involvement of DRD1 in this process, highlighting the intricate interplay between adenosine and dopamine receptors. The cAMP signaling pathway emerges as a critical mediator of the observed effects, with implications for cisplatin sensitivity enhancement and the reversal of cisplatin resistance in OC cells. These findings contribute to a deeper understanding of the molecular mechanisms underlying OC pathogenesis and may hold promise for the development of targeted therapeutic strategies.

Although the findings of the study suggested that the modulation of cisplatin's anticancer effect was influenced by the gut microbiota through metabolites originating in the gut, it is imperative to recognize several limitations. The results were derived from an OC mice model, where the gut microbiota was depleted using ABXs to investigate the involvement of commensal bacteria. Future research endeavors should prioritize the validation of the gut microbiota's role in a germ-free mouse model. Additionally, our findings indicated a disruption in caffeine metabolism due to ABX treatment, yet the specific underlying mechanism remains elusive. To gain further insights into the role of the gut microbiota in caffeine metabolism, it is worth exploring the alterations in gut lumen metabolomics in mice treated with bacteria that may play a role in caffeine metabolism. Alternatively, investigating the supernatant of chow co-cultured with feces from mice or specific bacteria could potentially shed light on this aspect.

In conclusion, our study presents compelling evidence of a cross-talk mechanism between gut microbiota and the anticancer efficacy of cisplatin in an OC mouse model. Furthermore, we have identified a key gut-derived metabolite, 3-methylxanthine, which enhances the response to cisplatin by promoting apoptosis through DRD1 activation(Fig. 7). This discovery sheds light on the remarkable role of the gut-ovary axis in regulating anticancer therapy and unveils potential therapeutic avenues for targeting cancer. The combination of cisplatin with gut-derived metabolites as adjuvants holds promising translational value in the context of anticancer therapy.

## MATERIALS AND METHODS

### Mice and tumor model

Six- to eight-week-old female C57BL/6 and BALB/c-nu mice, bred in SPF conditions, were obtained from Biotechnology Co., Ltd (Beijing, China). Germ-free mice were provided by Cyagen Biosciences (Suzhou, China). The mice were housed in a controlled environment with a 12 h light/dark cycle and provided standard rodent chow *ad libitum*.

### Cell lines and cell culture

The luciferase-tagged ID8 cell line (ID8-luc), commonly utilized for establishing a syngeneic mouse model of human OC, was cultured in DMEM (Gibco, Waltham, USA) supplemented with 10% fetal bovine serum (FBS) (Gibco, Waltham, USA), SKOV3, a human epithelial OC cell line, was cultured in McCoy's 5A medium (Gibco, Waltham, USA) supplemented with 10% FBS. All cell lines were gifts from Mading laboratory, Huazhong University of Science and Technology and were maintained at 37°C in a 5% $CO_2$ humidified atmosphere.

### Syngeneic tumor model

Female C57BL/6 mice, aged 6–8 weeks, were employed to establish an intraperitoneal tumor model by injecting ID8-luc cells, as previously described (54). Briefly, C57BL/6

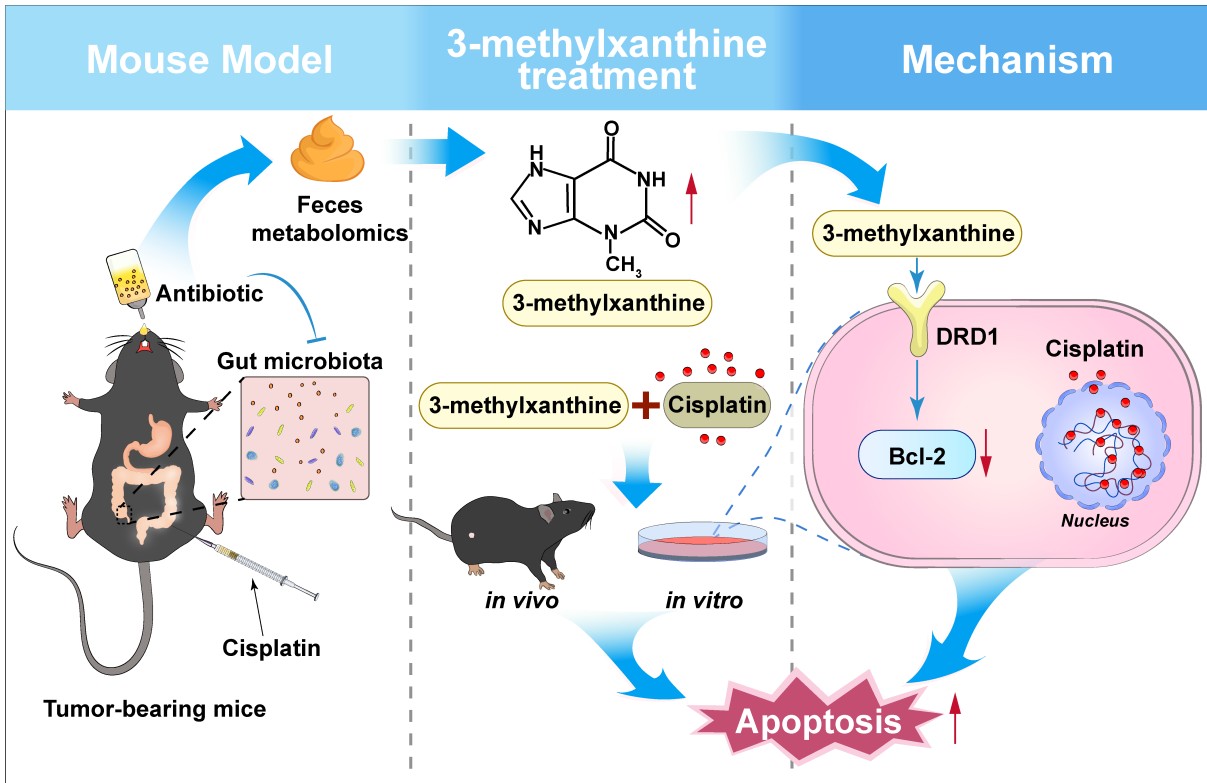

**FIG 7** The abstract graphic. Our study elucidated the exacerbating effect of ABX-induced gut microbiota depletion on the anticancer efficacy of cisplatin in a murine tumor-bearing model. Metabolomic analysis revealed a significant upregulation of 3-methylxanthine in mice treated with ABXs. Moreover, we demonstrated that 3-methylxanthine enhanced cisplatin-induced apoptosis both *in vivo* and *in vitro*. Mechanistically, 3-methylxanthine enhanced cisplatin-induced apoptosis through a pathway dependent on the DRD1-mediated modulation of the Bcl-2.

mice were intraperitoneally injected with $5 \times 10^6$ ID8-luc cells. Two weeks post-injection, tumor formation in the intraperitoneal xenograft tumor model was confirmed using BLI imaging. Tumor progression was monitored weekly using the BLI imaging system. At the end of the imaging period, the animals were humanely euthanized.

For the establishment of the subcutaneous xenograft tumor model, BALB/c-nu mice were used. $5 \times 10^6$ ID8-luc cells were subcutaneously injected into the right flank of each BALB/c-nu mouse. In this subcutaneous xenograft tumor model, tumor volumes were measured every 2 days using vernier calipers by assessing the length ($L$) and width ($W$) of the tumors. The tumor volume was calculated using the formula: tumor volume $= L \times W \times W/2$. Once the tumor size reached approximately 100 mm$^3$, the treatment interventions were initiated. The animals were euthanized when the tumors reached a volume of 800–1,000 mm$^3$, which served as the experimental endpoint. Subsequently, their samples were collected for further analyses and studies.

### *In vivo* BLI

BLI was conducted following established protocols for C57BL/6 mice injected with ID8-luc cells, as previously described (54). To initiate the imaging procedure, the animals were intraperitoneally injected with d-luciferin ( 150 mg/kg) (Selleck, Shanghai, China). After a 5–10 min interval, the mice were anesthetized using 2%–3% isoflurane, which was then reduced to 2% after transferring the animals to the imaging chamber.

For BLI measurements, a Bruker MI SE software imaging system (Bruker Corporation, Billerica, MA, USA) was utilized, and photon emission per second was recorded from each mouse at the optimal imaging time. The acquired BLI data were subsequently quantified using the Bruker MI SE software.

## Gut microbiota depletion and FMT

To deplete the gut microbiota in C57BL/6 mice, an ABX treatment regimen was implemented. The mice received an intragastric administration of a ABXs cocktail, including vancomycin (100 mg/kg; Macklin, Shanghai, China), neomycin sulfate (200 mg/kg; Macklin, Shanghai, China), metronidazole (200 mg/kg; Macklin, Shanghai, China), and ampicillin (200 mg/kg; Macklin, Shanghai, China). The ABX treatment was conducted once daily for five consecutive days before tumor implantation, and it was continued every 3 days until the completion of the experiment, following previously described procedures (55). The successful depletion of gut microbiota was confirmed by qPCR targeting the 16S rRNA gene in the cecum content of the mice.

In the FMT experiment, we collected feces from mice and resuspended in PBS at a concentration of 0.125 g/mL before ABX treatment. Following 5 days ABX treatment, the mice were intragastric received fecal suspension from mice before ABX treatment for 6 days. Feces were collected to evaluated the level of 3-methylxanthine by HPLC.

## Drug treatment

For drug treatment, cisplatin (Macklin, Shanghai, China) at a concentration of 5 mg/kg body weight was dissolved in PBS. In the experimental groups, mice were intraperitoneally injected with cisplatin every 5 days after tumor formation. Similarly, 3-methylxanthine (Macklin, Shanghai, China) at a concentration of 10 mg/kg body weight was dissolved in PBS, following previously described methods (28). In the designated groups, mice received daily intraperitoneal injections of 3-methylxanthine after tumor formation. Control groups of mice were administered with PBS using the same procedures as described above.

## Measurement of cytotoxicity

To evaluate the cytotoxic effects of 3-methylxanthine and cisplatin on ID8 and SKOV3 cells, 3-(4,5-dimethylthiazol-2-yl)-2,5-diphenyltetrazolium bromide (MTT, BeyoClickTM, China) assays were performed. Initially, the cells were seeded in 96-well plates at a density of $5 \times 10^3$ cells/well and allowed to adhere overnight. Subsequently, cisplatin (Macklin, Shanghai, China) was diluted to various concentrations using the corresponding complete medium and added to the wells, either with or without 2 mM 3-methylxanthine (Macklin, Shanghai, China). Following a 24 h incubation period, the culture media were replaced with a 0.5 mg/mL MTT solution. After 4 h, the MTT solutions were carefully removed, and 100 µL of dimethyl sulfoxide (DMSO) was added to each well to dissolve the formazan crystals. The plates were then subjected to gentle shaking for 10 min to ensure complete dissolution, and the absorbance was measured at 570 nm using a microplate reader (Epoch, Bio-Tek, USA).

## Cell proliferation assay

The EdU assay was conducted to evaluate cellular proliferation using the EdU kit (BeyoClick, EDU-488, China). A total of $5 \times 10^5$ cells/well were seeded in 12-well plates and incubated in the presence of 20 µM cisplatin, with or without 2 mM 3-methylxanthine, for 24 h. Subsequently, the cells were incubated with EdU for 2 h, fixed with 4% paraformaldehyde for 15 min, and permeabilized with 0.3% Triton X-100 for 15 min. The cells were then incubated with the click reaction mixture in the dark for 30 min at room temperature, followed by incubation with 4',6-diamidino-2-phenylindole (DAPI) for 10 min. Fluorescence microscopy (Olympus, Tokyo, Japan) was utilized for detection. The percentage of EdU-positive cells (EdU-positive/DAPI-positive) was quantified as the proportion of EdU-positive cells (%). Three independent experiments were performed to ensure robustness and reliability of the results.

## Flow cytometry

The annexin V-fluorescein isothiocyanate (FITC)/propidium iodide (PI) apoptosis kit (Multi sciences, Hangzhou, China) was employed to determine the percentage of apoptotic cells, following the manufacturer's instructions. A total of $2.5 \times 10^5$ cells were seeded into a 12-well dish. After overnight attachment, the cells were washed twice with PBS, and the medium was replaced with 20 µM cisplatin, 2 mM 3-methyl-xanthine, or 100 µM SCH-23390 hydrochloride (HY-19545A, MedChemExpress, USA). All cells, including the floating cells in the culture medium, were harvested. The cells were resuspended in 500 µL of ice-cold 5× binding buffer. Each cell suspension was mixed with 10 µL of FITC annexin V and 2 µL of PI. The mixture was incubated for 5 min at room temperature in the dark and subsequently analyzed using a Flow Cytometer (easyCyte, Luminex, USA). The obtained data were analyzed using FlowJo software (Tree Star, Inc., Ashland, OR).

## siRNA-mediated gene silencing

The siRNA sequences (Ribobio, Guangzhou, China) utilized in this study are provided as follows: 5′-ACCGATGTCTCTCTAGAAA-3′ (targeting DRD1). Cells were seeded and allowed to incubate overnight. Subsequently, the cells were transfected with the appropriate siRNA using Lipofectamine RNAiMAX Reagent (Invitrogen, Waltham, USA), following the manufacturer's instructions. The cells were cultured in medium supplemented with 10% FBS for 24 h before conducting downstream experiments.

## Gene expression analysis

Total RNA was isolated using Trizol reagent (Invitrogen, Carlsbad, USA), following the manufacturer's instructions. Subsequently, the cDNA was synthesized using a reverse transcription reagent kit (Toyobo, Osaka, Japan). The mRNA expressions were analyzed using SYBR Green Realtime PCR Master Mix (Toyobo, Osaka, Japan). qRT-PCR was conducted on an ABI 7500 real-time PCR system (Thermo Scientific, MA, USA). The gene expression levels were normalized to the endogenous control, 18S rRNA. The specific primers used for quantitative PCR were as follows: DRD1:Forward: 5′-TGGCAC AAGGCAAAACCTACA-3′, Reverse: 5′-CTGCTCAACCTCGTGTCACA-3′, 16S rRNA: Forward: 5′-GTGSTGCAYGGYTGTCGTCA-3′, Reverse: 5′-ACGTCRTCCMCACCTTCCTC-3′, 18S rRNA: Forward: 5′-CGATCCGAGGGCCTCACTA-3′, Reverse: 5′-AGTCCCTGCCCTTTGTACACA-3′. The relative expression levels of each target gene were calculated using the $2^{-\Delta\Delta Ct}$ method.

## Western blot

Protein extraction from cells was performed using a commercially available lysis buffer (Thermo, Rockford, USA). The resulting supernatants were collected through centrifugation at $12,000 \times g$ at 4°C for 15 min. The protein concentration was determined using a bicinchoninic acid (BCA) protein assay kit (Elabscience, Wuhan, China). Equal amounts of total protein were separated by 15% SDS-PAGE and subsequently transferred onto a polyvinylidene fluoride (PVDF) membrane (Millipore, Billerica, USA). The membrane was then blocked with 5% non-fat powdered milk (Biofroxx, Jena, Germany) for 1 h.

For immunoblotting, the membranes were first incubated overnight at 4°C with the respective primary antibodies. The primary antibodies used included gamma H2A.X (phospho S139) (γH2AX) (Abcam, Cambridgeshire, UK), pro-caspase 3 (Abcam, Cambridgeshire, UK), cleaved caspase-3 (Cell Signaling, MA, USA), Bcl-2 (Abcam, Cambridgeshire, UK), Bax (Proteintech, Wuhan, China), DRD1 (Abcam, Cambridgeshire, UK), and GAPDH (Abcam, Cambridgeshire, UK), which served as a protein loading control. Following primary antibody incubation, the membranes were probed with a horseradish peroxidase-conjugated secondary antibody (Proteintech, Wuhan, China). Protein antigen visualization was achieved using an enhanced chemiluminescent system, and the signals were recorded with Syngene Bio Imaging (Gene Gnome, Shanghai, China).

Image acquisition and densitometric analysis of the gels were performed using ImageJ software (https://imagej.nih.gov/ij).

## Biochemical analysis

Serum levels of Crea, ALT, and AST were quantified utilizing commercially available kits (Jiancheng Bioengineering, Nanjing, China) following the guidelines provided by the manufacturer.

## Histological examination and TUNEL staining

Tissue specimens were collected and subsequently fixed in 4% paraformaldehyde. The samples were then processed for embedding, sectioning, and subjected to H&E staining. TUNEL staining was conducted utilizing a TUNEL staining kit (Elabscience, Wuhan, China), following the recommended protocols provided by the manufacturer. Fluorescent images were captured using a fluorescence microscope (Olympus, Tokyo, Japan). To quantify apoptotic cells, the number of TUNEL-positive cells (green) and TUNEL-negative cells (DAPI) were counted in five random fields. The percentage of TUNEL-positive cells (TUNEL-positive/TUNEL-negative) was quantified as the proportion of TUNEL-positive cells (%).

## 16S rRNA sequencing

Cecum content samples obtained from mice at the experimental endpoint were promptly stored at −80°C. Fecal samples were homogenized in a PBS solution containing 0.5% Tween 20, followed by repeated cycles of freezing at −80°C for 10 min and thawing at 60°C for 5 min. Subsequently, DNA extraction and purification were carried out using the phenol-chloroform-isoamyl alcohol method and a commercially available kit (Solarbio, Beijing, China). The concentration and purity of the extracted DNA were determined using a NanoDrop spectrophotometer. For the amplification of the hypervariable region 4 (V4) of the bacterial 16S rRNA gene, PCR was performed with V4-F (5′-GTGTGYCAGCMGCCGCGGTAA-3′) and V4-R (5′-CCG GACTACNVGGGTWTC-TAAT-3′) primers. DNA sequencing was conducted on an Illumina HiSeq PE250 platform, and the resulting raw 16S rRNA gene sequencing data were processed and analyzed using the QIIME2 platform (version 2020.2).

## RNA sequencing

Tumors were harvested from mice, and total RNA was extracted from the samples using Trizol reagent (Invitrogen) following the manufacturer's instructions. RNA sequencing was conducted by Guangdong Magigene Biotechnology Co., Ltd. (Guangzhou, China). The quality of RNA, including degradation and contamination, was assessed using 1% agarose gel electrophoresis. The quantity of RNA was measured simultaneously using Qubit 3.0 and Nanodrop One). Furthermore, RNA integrity was accurately evaluated using the Agilent 4200 system (Agilent Technologies, Waldbron, Germany).

To generate whole mRNA sequencing libraries, the NEB Next Ultra Nondirectional RNA Library Prep Kit for Illumina (New England Biolabs, MA, USA) was employed according to the manufacturer's recommendations. In brief, mRNA was purified from the total RNA using poly-T oligo-attached magnetic beads. Fragmentation of the mRNA was carried out using NEB Next First Strand Synthesis Reaction Buffer. Subsequently, the first-strand cDNA was synthesized using a random hexamer primer and M-MuLV Reverse Transcriptase (RNase H), followed by second-strand cDNA synthesis using DNA polymerase I and RNase H. Remaining overhangs were then converted into blunt ends through exonuclease/polymerase activities. After adenylation of the 3′ ends of DNA fragments, NEB Next adaptors with a hairpin loop structure were ligated to prepare for hybridization. The cDNA fragments of preferentially 150–200 bp in length were selected using SpeedBead Magnetic Carboxylate Modified Particles (Global Life Sciences Solutions Operations, Buckinghamshire, UK). PCR was performed with Phusion High-Fidelity DNA

polymerase, Universal PCR primers, and Index (X) Primer. Finally, PCR products were purified using AMPure XP beads, and the library insert size was assessed on the Qsep400 High-Throughput Nucleic Acid Protein Analysis system (Houze Biological Technology Co, Hangzhou, China). The index-coded samples were clustered using a cBot Cluster Generation System, and the library was sequenced on an Illumina Novaseq 6000 platform, generating 150 bp paired-end reads.

Differential gene expression analysis was conducted using the R package DESeq2. PCA was performed on the transcriptome data using the R package vegan. GSEA was carried out using the R package clusterProfiler, with a threshold of $P < 0.05$ for GSEA results. Heatmaps and volcano plots were created using the R packages pheatmap and ggplot2, respectively. Correlation analysis was performed and visualized by corrplot package.

## High-performance liquid chromatography analysis

Feces were collected for the detection of 3-methylxanthine levels using high-performance liquid chromatography (Agilent 1260, CA, USA). Briefly, the standard 3-methylxanthine used in our study was supplied by Macklin, with a purity of 98%. We diluted the 3-methylxanthine in methanol to concentrations of 1, 7.8125, 15.625, 31.25, 62.5, 125, 250, 500, and 1,000 ng/mL. Subsequently, we analyzed these samples and plotted a standard curve based on the relationship between peak area and concentration. This standard curve allowed us to accurately determine the concentration of 3-methylxanthine in the samples.

All samples were extracted with methanol and centrifuged at $16,000 \times g$ for 15 min. The supernatant was processed by vacuum drying and reconstituted in 100 µL methanol. A total volume of 10 µL sample was injected into a Hypersil ODS (C18) column. For the detection of 3-methylxanthine, the liquid phase conditions were as follows: a mobile phase consisting of 0.1% formic acid water (A) and 0.1% formic acid acetonitrile (B). The gradient elution conditions were as follows: 0–0.5 min, 5% B; 0.5–10 min, 99% B; 9.0–9.1 min, 99%-2% B; 9.1–10 min, 5% B. The flow rate was 0.5 mL/min, and the column temperature was maintained at 40℃. Data were acquired and analyzed using Agilent LC1260 software.

## Metabolomic analysis by LC-MS

Metabolomic analysis was conducted by Guangdong Magigene Gene Technology Co., Ltd. Briefly, cecum contents collected from mice at the endpoint were promptly stored at −80℃ for preservation. Liquid chromatography-tandem mass spectrometry (LC-MS) analyses were performed using an UHPLC system (Vanquish, Thermo Fisher Scientific) with a UPLC BEH (2.1 mm × 100 mm, 1.7 µm) amide column coupled to Q Exactive HFX mass spectrometer (Orbitrap MS, Thermo). The mobile phase consisted of 25 mmol/L ammonium acetate and 25 ammonia hydroxide in water (pH = 9.75) (A) and acetonitrile (B). The auto-sampler temperature was 4℃, and the injection volume was 3 µL.

The Orbitrap Exploris 120 mass spectrometer was used for its ability to acquire MS/MS spectra on information-dependent acquisition (IDA) mode in the control of the acquisition software (Xcalibur, Thermo). In this mode, the acquisition software continuously evaluates the full scan MS spectrum. The electrospray ionization (ESI) source conditions were set as following: sheath gas flow rate as 50 Arb, Aux gas flow rate as 15 Arb, capillary temperature 320℃, full MS resolution as 60,000, MS/MS resolution as 15,000 collision energy as 10/30/60 in normalized collision energy (NCE) mode, spray voltage as 3.8 kV (positive) or −3.4 kV (negative), respectively.

## LC-MS data processing and analysis

The raw data underwent conversion to the mzXML format using ProteoWizard. Subsequently, we implemented an in-house program, developed in R based on XCMS, for the comprehensive tasks of peak detection, extraction, alignment, and integration (56).

Metabolite identification was performed following the methods outlined in previous literature (57). Furthermore, we optimized the utilization of our in-house MS2 database, BiotreeDB, encompassing both standard and publicly available components. Standards were meticulously generated through experimental procedures using the same instrumentation, and their MS/MS spectra were seamlessly incorporated into our database. The determination of identification levels now explicitly adheres to the criteria outlined in previous literature (58).

In cases where metabolites remained unidentified, we revisited our approach and turned to publicly available MS/MS spectral libraries, including Human Metabolome Database (HMDB) and Kyoto Encyclopedia of Genes and Genomes (KEGG). The matching criteria for both metabolic features and MS/MS spectra were refined to involve precise measures such as accurate masses (±25 ppm) and retention time values (±30 s). The annotation cutoff was judiciously set at 0.3.

To facilitate a more comprehensive analysis, we implemented several data management procedures on the original dataset. These steps primarily involved filtering individual peaks to eliminate noise, with the deviation value being filtered based on the relative standard deviation (RSD), also known as the coefficient of variation (CV). Additionally, we retained only peak area data with no more than 50% null values in one group or across all groups. Furthermore, we applied a technique to simulate missing values in the raw data, employing the numerical simulation method, specifically the minimum half method. Finally, as part of our data processing, we normalized the dataset using an internal standard (IS). The intensity data were log10 transformed and then scaled. Differential metabolites were classified based on the following criteria: (i) the $P$ value of Student's $t$-test was <0.05, (ii) the log2 fold change was >1.5 or <−1.5, and (iii) the variable importance in the projection (VIP) of the first principal component of the OPLS-DA model was above 1. KEGG enrichment analysis was performed by MetaboAnalyst 5.0 web platform. To predict the physiological function of 3-methylxanthine, we employed a modified strategy as previously described (59).

In the first arm of our study, we aimed to investigate functional and phenotypical alterations in tumor tissue by employing transcriptome sequencing in matched mice.

In the second arm, we aimed to build upon existing knowledge and explore the associations of 3-methylxanthine with other metabolites that share structural similarities. Firstly, metabolites with structural similarity to 3-methylxanthine were identified through a Fingerprint Tanimoto-based 2-dimensional similarity search in PubChem (https://pubchem.ncbi.nlm.nih.gov/), assuming that similar structural features imply similar biological functions. Subsequently, metabolite-associated genes were obtained from the STITCH database (Search Tool for Interactions of Chemicals, version 5.0), with a confidence score threshold of >0.9 indicating the highest level of confidence (60).

## Statistics

Statistical analyses were conducted using Prism 8.0 software (GraphPad Software, Inc., San Diego, CA, USA). Normality of the data were assessed before analysis. Differences among groups were evaluated using two-tailed, unpaired Student's $t$-test and Mann–Whitney $U$ test, one-way analysis of variance (ANOVA) with Dunnett's test, Kruskal–Wallis test and two-way ANOVA followed by Tukey's test for multiple comparisons, as appropriate. The results are presented as mean ± standard error of the mean (SEM). A $P$-value <0.05 was considered statistically significant. All $P$ values are indicated in the figures and figure legends (*$P < 0.05$; **$P < 0.01$; ***$P < 0.001$; ****$P < 0.0001$; ns, no significant difference).

## ACKNOWLEDGMENTS

We would like to express heartfelt gratitude to Jianping Xu and Huiling Shang for their encouragement and support.

This work was supported by the National Natural Science Foundation of China (8210063004 [X.C.], 82201812 [Z.M.]); the Natural Science Foundation of Guangdong

Province (2022A1515110278 [Z.M.], 2022A1515140128 [X.C.], 20211515110183 [Y.F.], and 2020A1515110321 [X.C.]), the Medical Scientific Research Foundation of Guangdong Province (A2022277 [X.C.]), and the China Postdoctoral Science Foundation (2022M710696 [Z.M.], 2023T160108 [Z.M.]). The funding bodies had no role in the study design, data collection or analysis, decision to publish, or preparation of the manuscript.

Y.W., X.C., Y.F., and Z.M. were responsible for the study design, supervision, and manuscript preparation; Z.M., Y.H., D.L., F.M., Y.L., Y.C., C.M., W.L., and S.Z. were responsible for the experiment; Z.M., H.Z., and P.L. were responsible for statistical and bioinformatic analyses. All authors provided critical revisions and approved the final manuscript.

## AUTHOR AFFILIATIONS

[1]Obstetrics and Gynecology Center, Zhujiang Hospital, Southern Medical University, Guangzhou, China

[2]Department of Obstetrics & Gynecology, First people's hospital of Foshan, Foshan, China

[3]Institute of Ecological Sciences, School of Life Sciences, South China Normal University, Guangzhou, China

[4]Department of Obstetrics and Gynecology, The First Affiliated Hospital of Ningbo University, Ningbo, Zhejiang Province, China

[5]Microbiome Research Centre, St. George and Sutherland Clinical School, UNSW, Sydney, New South Wales, Australia

[6]Department of Obstetrics, Affiliated Foshan Maternity & Child Healthcare Hospital, Southern Medical University, Foshan, China

## AUTHOR ORCIDs

Zhensheng Mai http://orcid.org/0000-0003-0642-240X
Huimin Zheng http://orcid.org/0000-0003-3489-0964
Yinglin Feng http://orcid.org/0009-0008-0980-0924
Xia Chen http://orcid.org/0000-0002-7915-4568
Yifeng Wang http://orcid.org/0000-0002-9559-7211

## FUNDING

| Funder | Grant(s) | Author(s) |
| --- | --- | --- |
| MOST \| National Natural Science Foundation of China (NSFC) | 8210063004 | Xia Chen |
| MOST \| National Natural Science Foundation of China (NSFC) | 82201812 | Hui-min Zheng |
| GDSTC \| Natural Science Foundation of Guangdong Province (廣東省自然科學基金) | 2022A1515110278 | Hui-min Zheng |
| GDSTC \| Natural Science Foundation of Guangdong Province (廣東省自然科學基金) | 2022A1515140128 | Xia Chen |
| GDSTC \| Natural Science Foundation of Guangdong Province (廣東省自然科學基金) | 2021A1515110183 | Yinglin Feng |
| GDSTC \| Natural Science Foundation of Guangdong Province (廣東省自然科學基金) | 2020A1515110321 | Xia Chen |
| Guangdong Medical Research Foundation (Guangdong Province Medical Research Foundation) | A2022277 | Xia Chen |
| China Postdoctoral Science Foundation (China Postdoctoral Foundation Project) | 2023T160108 | Hui-min Zheng |

## DATA AVAILABILITY

The data are included in the article as figures and tables, and other details can be obtained through email from corresponding author.The 16S rRNA-seq, RNA-Seq and metabolomics data are openly available in Chinese National Gene Bank Nucleotide Sequence Archive at (https://db.cngb.org/cnsa/), reference numbers CNP0004558, CNP0004602, and CNP0005441. The authors confirm that the data supporting the findings of this study are available within the article and its supplemental materials.

## ETHICS APPROVAL

The study protocols adhered to the guidelines set by the National Institutes of Health and were approved by the Animal Care and Use Committee of the First People's Hospital of Foshan (Approval No. 20221101 sr-mouse-1). Stringent measures were taken to ensure the welfare and ethical treatment of the animals throughout the experimental procedures.

## ADDITIONAL FILES

The following material is available online.

### Supplemental Material

**Supplemental Figures (mSystems01301-23-S0001.pdf).** Figures S1-S5.
**Supplemental Tables (mSystems01301-23-S0002.xlsx).** Tables S1-S4.

### Open Peer Review

**PEER REVIEW HISTORY (review-history.pdf).** An accounting of the reviewer comments and feedback.

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
