## [Reviewer comments · mSystems]

Gut-derived metabolite 3-methylxanthine enhances cisplatin-induced apoptosis via dopamine receptor D1 in a mouse model of ovarian cancer

Zhensheng Mai, Yubin Han, Dong Liang, Feihong Mai, Huimin Zheng, Pan Li, Yuan Li, Cong Ma, Yunqing Chen, Weifeng Li, Siyou Zhang, Yinglin Feng, Xia Chen, and Yinfeng Wang

Corresponding Author(s): Xia Chen, The First Affiliated Hospital of Ningbo University

Review Timeline:

Submission Date:	December 6, 2023
Editorial Decision:	February 14, 2024
Revision Received:	April 6, 2024
Editorial Decision:	April 24, 2024
Revision Received:	April 29, 2024
Accepted:	April 30, 2024

Editor: Neha Garg

Reviewer(s): Disclosure of reviewer identity is with reference to reviewer comments included in decision letter(s). The following individuals involved in review of your submission have agreed to reveal their identity: Elliot S Friedman (Reviewer #1)

Transaction Report:

DOI: <https://doi.org/10.1128/msystems.01301-23>

Re: mSystems01301-23 (**Gut-derived metabolite 3-methylxanthine enhances cisplatin-induced apoptosis via dopamine receptor D1 in a mouse model of ovarian cancer**)

Dear Mr. Xia Chen:

Revision Guidelines

Sincerely,
Neha Garg
Editor
mSystems

Reviewer #1 (Comments for the Author):

Figure 2C - I'm confused. What does the length of the gray bar next to each metabolite name indicate? Also, please change color scheme. Having the groups and also the log2FC be both blue is confusing.

3-methylxanthine is not upregulated. Levels are higher.

Figure 3A is not analysis of differential metabolites. These are pathways.

Does Figure 3B represent the 'Caffeine Metabolism' pathway in KEGG? If so, please use 'Caffeine Metabolism' instead of 'caffeine degradation'

I do not follow Figure 3C. what does 'metabolite with similar structure' indicate?

Using antibiotics is essentially a 'sledgehammer' in regards to the microbiota. Have the authors investigated which taxa are involved in caffeine/3-methylxanthine metabolism and specifically targeted these taxa with specific abx? Alternatively using GF mice w/ and w/o microbiota capable of producing 3-methylxanthine? Or mice colonized with a consortia of organisms w/ or w/o the taxa capable of producing 3-methylxanthine?

Flow cytometry is spelled incorrectly on line 616.

Please specify how chromatographic peaks were annotated to specific metabolites. Were these using standards? MS/MS spectra generated using the same instrument used in this study? Publicly available MS/MS spectra? What were the criteria for annotating/identifying peaks?

Were corrections made for multiple comparisons using false discovery rate (FDR)?

Figures is spelled incorrectly on line 796.

Please deposit metabolomics data and provide accession number(s).

Reviewer #2 (Comments for the Author):

In this manuscript, Mai et al discover that treatment of ovarian cancer-bearing mice with cisplatin is more effective when co-administered with antibiotics (Abx + Cisplatin). Through untargeted metabolomics, they determined that there are changes to the levels of 3-methylxanthine and related xanthine derivatives in the cecum of antibiotic-treated vs. untreated (OC-bearing cisplatin-treated) mice. Using multiomic analysis, the authors determine that there is a significant enrichment of cAMP signaling in Abx + Cisplatin treated mice and increased expression of apoptosis-related genes. They suggest that 3-methylxanthine and observed changes in xanthine metabolism is responsible for increased apoptosis and better outcomes in antibiotic-treated mice. This hypothesis was supported by subsequent in vivo studies which confirmed the synergistic effect of 3-methylxanthine with cisplatin in upregulating expression of DRD1. This study suggests that 3-methylxanthine could be a safe and effective way to boost the efficacy of cisplatin therapy through enhanced apoptosis.

Major comments:

- In the metabolomics analysis, there are many metabolites more upregulated in the Abx + cisplatin group than 3-methyl xanthine that are not mentioned by the authors. It would be helpful to include a short discussion of the metabolomics results beyond the xanthines.
- The authors state that increased levels of caffeine and thioguanine are detected in antibiotic-treated mice. Did the authors perform retention time and MS/MS matching with authentic standards to be certain that these are the correct annotations? It seems like the authors are proposing that these compounds can be synthesized de novo in mice, but the biosyntheses have not been reported prior (at least to this reviewer's knowledge).
- The authors should specify how they obtained annotations or library matches for their LC-MS/MS features in their methods as it is currently not listed.
- The authors propose that the gut microbiota are responsible for modulating caffeine metabolism. However, the mice in their study are not receiving caffeine so this should be referred to as purine metabolism. This reference (<https://doi.org/10.1016/j.chom.2023.05.011>) can be added in support of the author's hypothesis.
- Why were metabolites with structural similarity to ononetin identified using the Fingerprint Tanimoto-based two-dimensional similarity search? Why not search for metabolites with xanthine-like fragmentation? Please add an explanation in the methods to support why the authors chose ononetin for their search.
- The findings presented here are opposite to what was found previously in mice. In reference 11, restoration of the gut microbiome reduces tumor progression and antibiotic treatment decreases the efficacy of cisplatin. Could the authors please justify why their experiments led to differing conclusions by highlighting differences in experimental design, etc.?

Minor/editorial comments:

- Appropriate references should be added for line 106-107, line 233 (for GSEA), lines 249-252
- Acronyms should be defined at their first use - PBS (line 146), ABX (line 146), BLI (line 150), GSEA (line 233)
- Line 255 should be reworded as "treated with vehicle, 3-methyl xanthine + cisplatin or cisplatin alone".
- In lines 332-350, the text does not explicitly state that 3-methyl xanthine was administered in addition to cisplatin and instead refers several times to the apoptosis abilities of 3-methylxanthine. However, mice were always treated with cisplatin in these studies and the results should be discussed accordingly in this paragraph.
- There are some spelling errors that should be corrected on lines 17, 29, 428, 431, 436 and 616 as well as in Fig. 1c (fold change), Fig. 1f and 4e (Tunel positive)
- glycodeoxycholic acid should be lowercase in line 109

Response to Reviewer Comments

Reviewer 1

Question 1: Figure 2C - I'm confused. What does the length of the gray bar next to each metabolite name indicate? Also, please change color scheme. Having the groups and also the log₂FC be both blue is confusing.

Response: Thank you for your valuable feedback on Figure 2C. We have revised the figure legend on line 1142-1143 to clarify that the length of the gray bar next to each metabolite name represents variable importance in projection (VIP) scores. Additionally, we have modified the color scheme to enhance clarity: the cisplatin group is now depicted in a light red shade, while the cisplatin + ABX group is represented in a light blue shade.

Question 2: 3-methylxanthine is not upregulated. Levels are higher.

Response: We would like to express our gratitude for your thorough review and valuable comments on our work. Regarding your point about the expression levels of 3-methylxanthine, we appreciate your correction on the accuracy of our terminology. You are correct in noting that the term "upregulated" may not be the most accurate descriptor of the changes observed in 3-methylxanthine during our experiments. Indeed, what we have observed is a significant increase in the levels of this compound.

Following your suggestion, we have revised the relevant text in lines 208-209, 216, 223-226, 1139-1140, 1145 to more accurately reflect the experimental results.

Question 3: Figure 3A is not analysis of differential metabolites. These are pathways. Does Figure 3B represent the 'Caffeine Metabolism' pathway in KEGG? If so, please use 'Caffeine Metabolism' instead of 'caffeine degradation'

Response: Thank you for your insightful comment regarding Figure 3A. We appreciate your attention to detail. Upon reviewing your concern, we acknowledge that Figure 3A indeed represents pathway enrichment analysis rather than the analysis of differential metabolites. Our study utilized the capabilities of the MetaboAnalyst website, particularly focusing on the functionalities of Enrichment Analysis and Pathway Analysis. Our results in Figure 3A stem from Enrichment Analysis, specifically targeting metabolic pathways. Therefore, we concur with your suggestion to modify the title of Figure 3A to "Metabolic Pathway Enrichment Analysis." This adjustment accurately reflects the nature of the analysis performed and ensures clarity for the readers regarding the methodology employed..

We have carefully reviewed Figure 3B, and we confirm that it indeed represents the 'Caffeine Metabolism' pathway in KEGG. We appreciate your suggestion, and we have updated the caption to reflect the accurate terminology. 'Caffeine Metabolism' is now used instead of 'caffeine degradation' in lines 1146-1147.

Question 4: I do not follow Figure 3C. what does 'metabolite with similar structure' indicate?

Response: We assumed that metabolite share structural similarity might be involved in similar biological function(1, 2). We used PubChem to search for structurally similar

metabolites with 3-methylxanthine through a Fingerprint Tanimoto-based 2-dimensional similarity search and selected the top 10 relevant metabolites. 'metabolite with similar structure' are refer to the metabolite shared structural similarity with 3-methylxanthine and they were listed in supplementary table S2.

Question 5: Using antibiotics is essentially a 'sledgehammer' in regards to the microbiota. Have the authors investigated which taxa are involved in caffeine/3-methylxanthine metabolism and specifically targeted these taxa with specific abx? Alternatively using GF mice w/ and w/o microbiota capable of producing 3-methylxanthine? Or mice colonized with a consortia of organisms w/ or w/o the taxa capable of producing 3-methylxanthine?

Response: Thank you for your astute inquiries concerning our research on the interplay between the microbiota and caffeine/3-methylxanthine metabolism. We recognize the significance of identifying the specific taxa involved in caffeine/3-methylxanthine metabolism and the potential for employing targeted antibiotic treatments to modulate these taxa. Regarding your question, we must honestly acknowledge that our study did not investigate the taxa involved in caffeine/3-methylxanthine metabolism nor did we specifically target these taxa with specific antibiotics.

However, in light of your valuable suggestions, we have designed a experimental approach to further confirmed the relationship between the gut microbiota and the level of 3-methylxanthine in the gut lumen. Our study involved the use of six to

eight-week-old specific pathogen-free (SPF) female C57BL/6 mice and germ-free mice. First we detected the level of 3-methylxanthine in the feces between the SPF mice and germ free mice by HPLC (High Performance Liquid Chromatography). Secondly, The experimental protocol entailed a five-day course of an antibiotic cocktail (ABX) to deplete the gut microbiota in SPF mice. This was followed by a six-day fecal microbiota transplantation (FMT) regimen to evaluate the restoration of 3-methylxanthine metabolism. Feces were collected to evaluated the level of 3-methylxanthine by HPLC. The experimental procedure is illustrated in the following diagram (Fig.1 A).

Our preliminary findings reveal a notable elevation in 3-methylxanthine levels in germ-free mice when compared to their SPF counterparts (Fig.1 B). This observation underscores a pivotal role of the gut microbiota in the modulation of 3-methylxanthine levels. Subsequent to the ABX treatment, SPF mice exhibited an increased concentration of 3-methylxanthine, corroborating our metabolomic analyses that suggested a marked elevation of 3-methylxanthine. Notably, a reduction in 3-methylxanthine levels was discerned following 3 days and 6 days of FMT (Fig.1 C), further endorsing the contributory role of gut microbiota in the production of 3-methylxanthine. PCR demonstrated a significant alteration in bacterial load in mice following administration of ABX and FMT (Fig.1 D)

Figure.1 (A) The level of 3-methylxanthine in the feces of mice was quantified using HPLC. (B) The level of 3-methylxanthine in the feces between in the germ free SPF mice. (n=6). (C) Feces were collected after ABX treatment and FMT. The level of 3-methylxanthine in the feces was measured by HPLC. (n=6). (D) Total fecal bacterial load. (n=5). Ctrl: Feces collected before ABX treatment; ABX: Feces collected after ABX treatment; FMT: Feces collected 3 days after FMT. FMT2: Feces collected 6 days after FMT. Error bars represent SEM. (A)* $p < 0.05$; ** $p < 0.01$, **** $p < 0.0001$.

These findings elucidate the influential role of the gut microbiota in the production of 3-methylxanthine and underscore the necessity for further investigations to pinpoint the specific microbial taxa implicated in this metabolic process.

In response to the your comments, we have made several revisions to the manuscript. Specifically, we have updated the Methods section as follows: added information on the source of germ free mice at lines 565-566, included details of the FMT method at lines 621-625 and provided specific parameters for HPLC at lines 787-797. Additionally, we have relocated the Results section to lines 254-268 and added a discussion section at lines 471-474.

Question 6: Flow cytometry is spelled incorrectly on line 616.

Response: Thank you for your thorough review of our manuscript. We appreciate your attention to detail. We have addressed the spelling error, and the correct term "flow cytometry" has been updated on line 663.

Question 7: Please specify how chromatographic peaks were annotated to specific metabolites. Were these using standards? MS/MS spectra generated using the same instrument used in this study? Publicly available MS/MS spectra? What were the criteria for annotating/identifying peaks?

Response: We extend our heartfelt appreciation for your constructive feedback and insightful inquiries, which have greatly enriched our manuscript. Your meticulous review has been invaluable.

In response to your queries regarding chromatographic peak annotation, we wish to highlight modifications made to the relevant section:

The raw data underwent conversion to the mzXML format using ProteoWizard. Subsequently, we implemented an in-house program, developed in R based on XCMS, for the comprehensive tasks of peak detection, extraction, alignment, and integration(3).

Metabolite identification was performed following the methods outlined in previous literature(4). Furthermore, we optimized the utilization of our in-house MS2 database, BiotreeDB, encompassing both standard and publicly available components. Standards were meticulously generated through experimental procedures using the same instrumentation, and their MS/MS spectra were seamlessly incorporated into our database. The determination of identification levels now explicitly adheres to the criteria outlined in previous literature(5).

In cases where metabolites remained unidentified, we revisited our approach and turned to publicly available MS/MS spectral libraries, including HMDB (Human

Metabolome Database) and KEGG (Kyoto Encyclopedia of Genes and Genomes). The matching criteria for both metabolic features and MS/MS spectra were refined to involve precise measures such as accurate masses (± 25 ppm) and retention time values (± 30 s). The annotation cutoff was judiciously set at 0.3.

We trust that these modifications and clarifications in line 817-833 contribute to a more robust and comprehensible presentation of our methodology. Thank you once again for your dedicated time and insightful consideration of our work.

Question 8: Were corrections made for multiple comparisons using false discovery rate (FDR)?

Response: Thank you very much for your thorough review of our research and for providing valuable feedback. Your concerns regarding our decision not to use FDR correction in the differential metabolite selection are indeed crucial and merit detailed discussion.

In our study, we employed multiple criteria for the selection of differential metabolites, including a the P value of Student's t-test was less than 0.05, a log₂ fold change greater than 1.5 or less than -1.5, and a Variable Importance in Projection (VIP) value greater than 1 for the first principal component in the OPLS-DA model. These criteria were chosen after careful consideration, taking into account both previous literature reports and the specific circumstances of our experimental design(6-8).

Regarding the decision not to use FDR correction, we would like to provide a comprehensive explanation of our considerations. Firstly, we acknowledge that FDR

correction is an effective method for controlling multiple comparison issues. However, given the high-dimensional nature of our data, we were concerned that FDR correction might overly restrict the overall results. 3-methylxanthine in our study, after correction, exhibits a *P*-value of 0.08. It is noteworthy that some studies opt for a more lenient *P*-value threshold to mitigate the risk of overlooking potentially significant results(9). Hence, we opted for a *P*-value of Student's t-test threshold of 0.05, based on a nuanced understanding of the field and a thoughtful analysis of our experimental conditions.

Secondly, to enhance the biological relevance of our selection, we introduced additional criteria such as log₂ fold change and VIP values. This integrative approach aimed to balance the leniency of the *P*-value threshold while reducing potential errors and maintaining sensitivity to biological relevance.

While we understand the significance of FDR correction, we believe that our multidimensional approach with various criteria better aligns with the practicalities of our study.

Question 9: Figures is spelled incorrectly on line 796.

Response: Thank you for catching that error. We have addressed the spelling mistake, and "Figures" has been corrected on line 872.

Question 10: Please deposit metabolomics data and provide accession number(s).

Response: We appreciate your suggestion regarding the deposition of metabolomics data. The metabolomics data are now openly available in the Chinese National Gene Bank Nucleotide Sequence Archive at [<https://db.cngb.org/cnsa/>], and the reference

number is CNP0005441. We have also updated the information in the 'Data Availability' section of the manuscript to reflect this.

Reviewer 2

In this manuscript, Mai et al discover that treatment of ovarian cancer-bearing mice with cisplatin is more effective when co-administered with antibiotics (Abx + Cisplatin). Through untargeted metabolomics, they determined that there are changes to the levels of 3-methylxanthine and related xanthine derivatives in the cecum of antibiotic-treated vs. untreated (OC-bearing cisplatin-treated) mice. Using multiomic analysis, the authors determine that there is a significant enrichment of cAMP signaling in Abx + Cisplatin treated mice and increased expression of apoptosis-related genes. They suggest that 3-methylxanthine and observed changes in xanthine metabolism is responsible for increased apoptosis and better outcomes in antibiotic-treated mice. This hypothesis was supported by subsequent in vivo studies which confirmed the synergistic effect of 3-methylxanthine with cisplatin in upregulating expression of DRD1. This study suggests that 3-methylxanthine could be a safe and effective way to boost the efficacy of cisplatin therapy through enhanced apoptosis.

Major comments:

Question 1: In the metabolomics analysis, there are many metabolites more upregulated in the Abx + cisplatin group than 3-methylxanthine that are not mentioned

by the authors. It would be helpful to include a short discussion of the metabolomics results beyond the xanthenes.

Response: Thank you for your thorough attention to our research. Regarding the issue you raised about several metabolites showing elevated levels compared to 3-methylxanthine, such as Thioguanine, Genistein, 5,7-Dihydroxyisoflavone, and Ampicillin, it's worth noting that these metabolites, such as Thioguanine, Genistein, have been reported to have apoptotic-inducing effects in the literature(10, 11).

Firstly, the decision to focus on 3-methylxanthine was based on multiple considerations, with a primary factor being its significant upregulation in the Abx + cisplatin treatment group. 3-methylxanthine is an important metabolite in the caffeine metabolism pathway and is among the differential metabolites enriched in this pathway. Additionally, through the functional prediction of metabolites, 3-methylxanthine and its structurally similar metabolites have been found to interact with genes related to apoptosis, further supporting the emphasis on 3-methylxanthine.

While other elevated metabolites, such as Thioguanine, Genistein, have been reported in the literature to induce apoptosis, our study's primary objective is not to disregard the potential roles of these important metabolites. In the discussion section (lines 433-444), we will elaborate on the scientific rationale behind choosing 3-methylxanthine.

Through this discussion, we aim to clearly articulate the scientific rationale of our study, simultaneously respecting and briefly acknowledging the potential biological significance of other metabolites. Thank you once again for your valuable guidance.

Question 2: The authors state that increased levels of caffeine and thioguanine are detected in antibiotic-treated mice. Did the authors perform retention time and MS/MS matching with authentic standards to be certain that these are the correct annotations? It seems like the authors are proposing that these compounds can be synthesized de novo in mice, but the biosyntheses have not been reported prior (at least to this reviewer's knowledge).

Response: Thank you for your attention to our research and for raising important questions. Regarding your concerns, we indeed did not conduct further targeted mass spectrometry validation for the precise annotation of caffeine and thioguanine. We acknowledge this point and appreciate your correction. While our study provides initial annotations of metabolites at the metabolomics level, we understand that it is not sufficient to confirm the exact identity of these metabolites.

In addition, concerning the question of whether these two metabolites could potentially be synthesized de novo in mice, we currently lack direct evidence to suggest that they can be synthesized within the mouse system. However, there is evidence indicating that the gut microbiota are involved in the metabolism of caffeine(12) and thioguanine(13). This interaction between gut microbiota and these metabolites presents a potential area of research that could illuminate novel metabolic pathways and mechanisms. Understanding the extent to which gut microbiota contribute to the presence and modification of such metabolites may reveal new insights into

host-microbe interactions, potentially impacting therapeutic strategies and our understanding of metabolic health.

Question 3: The authors should specify how they obtained annotations or library matches for their LC-MS/MS features in their methods as it is currently not listed.

Response: We want to extend our sincere thanks for your constructive feedback and thoughtful inquiries, which have truly elevated the quality of our manuscript.

In response to your queries regarding chromatographic peak annotation, we wish to highlight modifications made to the relevant section:

The raw data underwent conversion to the mzXML format using ProteoWizard. Subsequently, we implemented an in-house program, developed in R based on XCMS, for the comprehensive tasks of peak detection, extraction, alignment, and integration(3).

Metabolite identification was performed following the methods outlined in previous literature(4). Furthermore, we optimized the utilization of our in-house MS2 database, BiotreeDB, encompassing both standard and publicly available components. Standards were meticulously generated through experimental procedures using the same instrumentation, and their MS/MS spectra were seamlessly incorporated into our database. The determination of identification levels now explicitly adheres to the criteria outlined in previous literature(5).

In cases where metabolites remained unidentified, we revisited our approach and turned to publicly available MS/MS spectral libraries, including HMDB (Human

Metabolome Database) and KEGG (Kyoto Encyclopedia of Genes and Genomes). The matching criteria for both metabolic features and MS/MS spectra were refined to involve precise measures such as accurate masses (± 25 ppm) and retention time values (± 30 s). The annotation cutoff was judiciously set at 0.3.

In lines 817-833, we provided a detailed description of our methodology for metabolite annotation. We believe that the adjustments and clarifications made will enhance the robustness and clarity of our methodology presentation.

Question 4: The authors propose that the gut microbiota are responsible for modulating caffeine metabolism. However, the mice in their study are not receiving caffeine so this should be referred to as purine metabolism. This reference (<https://doi.org/10.1016/j.chom.2023.05.011>) can be added in support of the author's hypothesis.

Response: To further elaborate on the key points you raised, we are willing to provide additional insights into the background, methods, and findings of our study in order to comprehensively address your comments.

Through in-depth metabolomics analysis, we observed a significant decrease in the concentrations of metabolites related to purine metabolism, such as xanthine, xanthosine, inosine, cAMP, and cGMP in cisplatin+ABX group (Table. S1). This trend aligns with observations in germ-free mice, further supporting the conclusion that gut microbiota may intervene in purine metabolism(14). However, we acknowledge that in this context, a more accurate description would be purine metabolism rather than caffeine metabolism, better reflecting the focus of our study. We extend our gratitude to

the reviewer for providing relevant references, particularly the one (<https://doi.org/10.1016/j.chom.2023.05.011>), which robustly supports our hypothesis regarding gut microbiota modulation of purine metabolism.

Our hypothesis that gut microbiota mediate intestinal caffeine metabolism is based on the discovery that, following antibiotic-induced gut microbiota depletion in mice, differential metabolites significantly enriched in the caffeine metabolism pathway. Additionally, we observed significant differences in caffeine metabolism pathway-related metabolites in the ABX+Cisplatin group. These findings further reinforce the potential regulatory role of gut microbiota in the caffeine metabolism pathway.

Finally, it's noteworthy that both caffeine metabolism and purine metabolism share the intermediate metabolite xanthine, suggesting a possible convergence in metabolic pathways. This observation opens up avenues for exploring how gut microbiota might impact both purine and caffeine metabolism. Further research is warranted to elucidate the intricate relationship between gut microbiota and the metabolism of caffeine and purines, enhancing our overall understanding in this area.

In lines 476-485, we integrated further discussions into the discussion section, building upon your valuable feedback.

Question 5: Why were metabolites with structural similarity to ononetin identified using the Fingerprint Tanimoto-based two-dimensional similarity search? Why not

search for metabolites with xanthine-like fragmentation? Please add an explanation in the methods to support why the authors chose ononetin for their search.

Response: Thank you for your review and valuable feedback. We acknowledge an error in the manuscript where "ononetin" was incorrectly used instead of "3-methylxanthine." We sincerely apologize for this mistake. Regarding your question about why we employed the Fingerprint Tanimoto-based two-dimensional similarity search to identify metabolites with structural similarity to 3-methylxanthine, rather than searching for metabolites with xanthine-like fragmentation, we made an error in our choice of words. The selection of ononetin for the search was a serious misdirection and does not align with the actual focus of our study.

Thank you once again for pointing out this issue. We will promptly make the necessary corrections in line 854 and enhance our research report with a more accurate method description.

Question 6: The findings presented here are opposite to what was found previously in mice. In reference 11, restoration of the gut microbiome reduces tumor progression and antibiotic treatment decreases the efficacy of cisplatin. Could the authors please justify why their experiments led to differing conclusions by highlighting differences in experimental design, etc.?

Response: Thank you for your thorough evaluation, and we appreciate the opportunity to address your concerns regarding the discrepancies between our study

results and those reported in reference 11. We have carefully considered the key points you raised and made detailed revisions to enhance the clarity of our manuscript.

Firstly, addressing the difference in antibiotic treatment plans, we have expanded our discussion on the impact of antibiotic selection and treatment strategies. While antibiotics were utilized in both studies to deplete the gut microbiota, previous studies have demonstrated that various antibiotic schemes have differing effects on the eradication of gut microbiota, which in turn have shown different outcomes in the progression of ovarian cancer(15, 16). Despite the challenges presented by the diversity of the microbiota and the differences it causes in research, antibiotic treatment remains an important tool for preliminary exploration of the impact of gut microbiota on diseases.

Secondly, we are emphasizing our use of in vivo imaging rather than evaluating tumor load through abdominal tumor volume assessment by ultrasound. We have clarified the differences in the timing of interventions and assessments and the cisplatin regimen. Through these revisions, we aim to make the manuscript clearer and more comprehensively reflect our considerations in experimental design, methodology, and temporal factors.

Additionally, we acknowledge the need for deeper research, including the identification of key microbial species and the study of crucial metabolites. These in-depth studies can help us explore the overall impact of gut microbiota on health and disease, especially how they play a role in the development and treatment of diseases.

This integrative approach might reveal new therapeutic targets, providing a scientific basis for developing new treatment methods.

Furthermore, in lines 451-461 of the Discussion section, we have supplemented our discussion to address these points accordingly.

Minor/editorial comments:

Question 7: Appropriate references should be added for line 106-107, line 233 (for GSEA), lines 249-252

Response: First and foremost, we extend our heartfelt gratitude for your valuable comments and suggestions. In response to the points you have raised, we have meticulously reviewed and accordingly updated our manuscript. Below are our replies to the specific suggestions you made:

Regarding the content in lines 109-110, we appreciate your pointing out the necessity of adding references. For this purpose, we have now included the following references: Le HH, et al. (2022); Xue H, et al. (2022); and Che Y, et al. (2023)(17-19). These additional references provide a solid scientific basis for our discussion on the impact of gut microbial metabolites in distant organ, enhancing the accuracy and depth of our manuscript.

In the discussion on Gene Set Enrichment Analysis (GSEA) at line 240-241, we have cited the study by Subramanian, A., et al. (2005)(20). This citation offers readers the original research and methodological background on the GSEA method, ensuring the accuracy and completeness of our methodological section.

For the content in lines 271-274, we have referenced the study by Nomura T. (1983)(21). This citation highlights the role of methylxanthine compounds (such as 3-methylxanthine) in inhibiting tumor occurrence in mice, providing additional experimental support for the mechanisms discussed in our study.

Question 8: Acronyms should be defined at their first use - PBS (line 146), ABX (line 146), BLI (line 150), GSEA (line 233)

Response: Thank you for your insightful review. We have addressed the concern about acronym definitions, and they are now defined at their first use in the manuscript. Specifically, we have provided definitions for PBS (line 150), ABX (line 151), BLI (line 155), and GSEA (line 241).

Question 9: Line 255 should be reworded as "treated with vehicle, 3-methylxanthine + cisplatin or cisplatin alone".

Response: Thank you for your thoughtful feedback. We have revised Line 277-278 in accordance with your suggestion. It now reads as follows: "PBS, 3-methylxanthine, cisplatin or cisplatin+3-methylxanthine treatment."

Question 10: In lines 332-350, the text does not explicitly state that 3-methylxanthine was administered in addition to cisplatin and instead refers several times to the apoptosis abilities of 3-methylxanthine. However, mice were always treated with cisplatin in these studies and the results should be discussed accordingly in this paragraph.

Response: Thank you very much for your careful reading of our manuscript and for bringing this particular issue to our attention. We understand your concern regarding the clarity of our discussion on the combined effects of 3-methylxanthine and cisplatin

in lines 332-350. We apologize for any confusion caused by our initial presentation and appreciate the opportunity to clarify this matter.

Upon reviewing the section in question, we recognize that our description might have inadvertently suggested that the apoptosis-inducing abilities of 3-methylxanthine were evaluated in isolation. We acknowledge that this was a significant oversight on our part. In all referenced studies, 3-methylxanthine was indeed administered in conjunction with cisplatin, and it is this combination treatment that was evaluated for its effects on inducing apoptosis in cancer cells.

To rectify this, we have revised lines 358-377 to explicitly state that 3-methylxanthine was used in addition to cisplatin in these studies. The revised text now clearly emphasizes that the enhanced apoptosis observed in the studies was a result of the combined treatment regimen, rather than the effect of 3-methylxanthine alone. We have made sure to detail the synergistic effect of 3-methylxanthine when administered alongside cisplatin, highlighting how this combination enhances the efficacy of cisplatin-induced apoptosis in cancer cells.

We hope that these revisions adequately address your concerns and clarify the intended message of this section. We are grateful for your insightful comments, which have undoubtedly improved the accuracy and clarity of our manuscript.

Question 11: There are some spelling errors that should be corrected on lines 20, 33, 428, 431, 436 and 616 as well as in Fig. 1c (fold change), Fig. 1f and 4e (Tunel positive)

Response: Thank you for your careful review of our manuscript. We have addressed the spelling errors as per your feedback. Corrections have been made on lines 17, 29, 462, 465, 470, and 663. Additionally, the spelling has been rectified in Fig. 1C (fold change), Fig. 1F, and Fig. 4E (TUNEL positive).

Question 12: glycodeoxycholic acid should be lowercase in line 109

Response: Thank you for your meticulous review of our manuscript. We appreciate your keen observation. The correction has been implemented, and "glycodeoxycholic acid" is now in lowercase on line 113.

Reference

1. Abdullah-Zawawi M-R, Govender N, Karim MB, Altaf-Ul-Amin M, Kanaya S, Mohamed-Hussein Z-A. 2022. Chemoinformatics-driven classification of Angiosperms using sulfur-containing compounds and machine learning algorithm. *Plant Methods* 18:118.
2. Altaf-Ul-Amin M, Katsuragi T, Sato T, Ono N, Kanaya S. 2014. An unsupervised approach to predict functional relations between genes based on expression data. *BioMed Research International* 2014:154594.
3. Smith CA, Want EJ, O'Maille G, Abagyan R, Siuzdak G. 2006. XCMS: processing mass spectrometry data for metabolite profiling using nonlinear peak alignment, matching, and identification. *Analytical Chemistry* 78:779-787.
4. Liang L, Rasmussen M-LH, Piening B, Shen X, Chen S, Röst H, Snyder JK, Tibshirani R, Skotte L, Lee NC, Contrepolis K, Feenstra B, Zackriah H, Snyder M, Melbye M. 2020. Metabolic Dynamics and Prediction of Gestational Age and Time to Delivery in Pregnant Women. *Cell* 181.
5. Alseekh S, Aharoni A, Brotman Y, Contrepolis K, D'Auria J, Ewald J, C Ewald J, Fraser PD, Giavalisco P, Hall RD, Heinemann M, Link H, Luo J, Neumann S, Nielsen J, Perez de Souza L, Saito K, Sauer U, Schroeder FC, Schuster S, Siuzdak G, Skirycz A, Sumner LW, Snyder MP, Tang H, Tohge T, Wang Y, Wen W, Wu S, Xu G, Zamboni N, Fernie AR. 2021. Mass spectrometry-based metabolomics: a guide for annotation, quantification and best reporting practices. *Nature Methods* 18:747-756.
6. Li Y, Zhao H, Sun G, Duan Y, Guo Y, Xie L, Ding X. 2022. Alterations in the gut microbiome and metabolome profiles of septic rats treated with aminophylline. *Journal of Translational Medicine* 20:69.

7. Fu M, Zhang X, Liang Y, Lin S, Qian W, Fan S. 2020. Alterations in Vaginal Microbiota and Associated Metabolome in Women with Recurrent Implantation Failure. *MBio* 11.
8. Oliveira MS, Santo RCE, Silva JMS, Alabarse PVG, Brenol CV, Young SP, Xavier RM. 2023. Urinary metabolomic biomarker candidates for skeletal muscle wasting in patients with rheumatoid arthritis. *Journal of Cachexia, Sarcopenia and Muscle* 14:1657-1669.
9. Varkey A, Devi S, Mukhopadhyay A, Kamat NG, Pauline M, Dharmar M, Holt RR, Allen LH, Thomas T, Keen CL, Kurpad AV. 2020. Metabolome and microbiome alterations related to short-term feeding of a micronutrient-fortified, high-quality legume protein-based food product to stunted school age children: A randomized controlled pilot trial. *Clinical Nutrition (Edinburgh, Scotland)* 39:3251-3261.
10. Zhang D, An X, Li Q, Man X, Chu M, Li H, Zhang N, Dai X, Yu H, Li Z. 2020. Thioguanine Induces Apoptosis in Triple-Negative Breast Cancer by Regulating PI3K-AKT Pathway. *Frontiers In Oncology* 10:524922.
11. Dev A, Sardoiwala MN, Kushwaha AC, Karmakar S, Choudhury SR. 2021. Genistein nanoformulation promotes selective apoptosis in oral squamous cell carcinoma through repression of 3PK-EZH2 signalling pathway. *Phytomedicine : International Journal of Phytotherapy and Phytopharmacology* 80:153386.
12. Ceja-Navarro JA, Vega FE, Karaoz U, Hao Z, Jenkins S, Lim HC, Kosina P, Infante F, Northen TR, Brodie EL. 2015. Gut microbiota mediate caffeine detoxification in the primary insect pest of coffee. *Nature Communications* 6:7618.
13. Oancea I, Movva R, Das I, Aguirre de Cárcer D, Schreiber V, Yang Y, Purdon A, Harrington B, Proctor M, Wang R, Sheng Y, Lobb M, Lourie R, Ó Cuív P, Duley JA, Begun J, Florin THJ. 2017. Colonic microbiota can promote rapid local improvement of murine colitis by thioguanine independently of T lymphocytes and host metabolism. *Gut* 66:59-69.
14. Kasahara K, Kerby RL, Zhang Q, Pradhan M, Mehrabian M, Lusic AJ, Bergström G, Bäckhed F, Rey FE. 2023. Gut bacterial metabolism contributes to host global purine homeostasis. *Cell Host & Microbe* 31.
15. Chambers LM, Esakov Rhoades EL, Bharti R, Braley C, Tewari S, Trestan L, Alali Z, Bayik D, Lathia JD, Sangwan N, Bazeley P, Joehlin-Price AS, Wang Z, Dutta S, Dwidar M, Hajjar A, Ahern PP, Claesen J, Rose P, Vargas R, Brown JM, Michener CM, Reizes O. 2022. Disruption of the Gut Microbiota Confers Cisplatin Resistance in Epithelial Ovarian Cancer. *Cancer Research* 82:4654-4669.
16. Chen L, Zhai Y, Wang Y, Fearon ER, Núñez G, Inohara N, Cho KR. 2021. Altering the Microbiome Inhibits Tumorigenesis in a Mouse Model of Oviductal High-Grade Serous Carcinoma. *Cancer Research* 81:3309-3318.
17. Le HH, Lee M-T, Besler KR, Johnson EL. 2022. Host hepatic metabolism is modulated by gut microbiota-derived sphingolipids. *Cell Host & Microbe* 30.
18. Xue H, Chen X, Yu C, Deng Y, Zhang Y, Chen S, Chen X, Chen K, Yang Y, Ling W. 2022. Gut Microbially Produced Indole-3-Propionic Acid Inhibits Atherosclerosis by Promoting Reverse Cholesterol Transport and Its Deficiency Is Causally Related to Atherosclerotic Cardiovascular Disease. *Circulation Research* 131:404-420.
19. Che Y, Chen G, Guo Q, Duan Y, Feng H, Xia Q. 2023. Gut microbial metabolite butyrate improves anticancer therapy by regulating intracellular calcium homeostasis. *Hepatology (Baltimore, Md)* 78.

20. Subramanian A, Tamayo P, Mootha VK, Mukherjee S, Ebert BL, Gillette MA, Paulovich A, Pomeroy SL, Golub TR, Lander ES, Mesirov JP. 2005. Gene set enrichment analysis: a knowledge-based approach for interpreting genome-wide expression profiles. *Proceedings of the National Academy of Sciences of the United States of America* 102:15545-15550.
21. Nomura T. 1983. Comparative inhibiting effects of methylxanthines on urethan-induced tumors, malformations, and presumed somatic mutations in mice. *Cancer Research* 43:1342-1346.

Re: mSystems01301-23R1 (**Gut-derived metabolite 3-methylxanthine enhances cisplatin-induced apoptosis via dopamine receptor D1 in a mouse model of ovarian cancer**)

Dear Mr. Xia Chen:

Thank you for the privilege of reviewing your work. Your manuscript is close to final acceptance. Reviewer 2 has signed off for acceptance and reviewer have pending minor modifications to be addressed. Below you will find instructions from the mSystems editorial office, and the reviewer comments.

Revision Guidelines

Sincerely,
Neha Garg
Editor
mSystems

Reviewer #1 (Comments for the Author):

There are multiple grammatical and spelling errors throughout the manuscript. Please fix them prior to publication (e.g., line 1145 "toal" instead of 'total', line 1142 "micewas" instead of 'mice was')

I appreciate that the authors have added a methods section regarding the quantification of 3-methylxanthine by HPLC. However, this section lacks information on the standard (e.g., purity, vendor) and the methods used for quantification against a standard (e.g., standard curve - how many points, replicates, range of assay, etc).

The legend to figure 2B is confusing. Specifically, the colors indicate 'up' or 'down' but the notation indicates enriched in cisplatin or ABX+cisplatin mice. Please revise for consistency.

Figure 2C is still confusing to me. With the exception of the VIP score, this shows exactly the same information as Figure 2B. Why show both? This is unnecessarily confusing to readers of your manuscript.

Figure 3A is confusing to me. The figure legend says that "the colors indicate the change of metabolites in the ABX+cisplatin group" but the legend within Figure 3A. Says that the colors indicate p-value. Which is it?

Figure 3B is labeled as 'the caffeine metabolism pathway'. However, this figure only shows a select portion of the caffeine metabolism pathway in KEGG (https://www.genome.jp/kegg-bin/show_pathway?map00232). Please elaborate on what this is, and why only specific portions of the pathway have been shown.

Figure legend of figure 3F is not accurate. It says "The level of 3-methylxanthine in the feces of micewas quantified using HPLC." However, the figure is of an experimental design.

Please show individual values in all figures currently showed as column/bar charts (e.g., 1C, 1D, 1E, 3B, 3G-3I, 4B-4C, 4E, etc). Showing column plots with SEM suggest that the standard deviation is high.

Reviewer #2 (Comments for the Author):

Thank you for implementing the comments and feedback provided in my review.

Response to Reviewer Comments

Reviewer 1

Question 1: There are multiple grammatical and spelling errors throughout the manuscript. Please fix them prior to publication (e.g., line 1145 "toal" instead of 'total', line 1142 "micewas' instead of 'mice was')

Response: Thank you for your valuable feedback. We have carefully reviewed the manuscript and addressed the grammatical and spelling errors you pointed out. Specifically, we have corrected the errors in lines 158, 255, 364, 465, 1152, 1161, 1174-1176, 1186, and 1204. We appreciate your thorough review and attention to detail.

Question 2: I appreciate that the authors have added a methods section regarding the quantification of 3-methylxanthine by HPLC. However, this section lacks information on the standard (e.g., purity, vendor) and the methods used for quantification against a standard (e.g., standard curve - how many points, replicates, range of assay, etc).

Response: We appreciate your suggestion and have made the necessary revisions to the manuscript to include more details regarding the quantification of 3-methylxanthine.

The standard 3-methylxanthine used in our study was supplied by Macklin, with a purity of 98%. For quantification, we employed high-performance liquid chromatography (HPLC) and established a standard curve. Initially, we diluted the 3-methylxanthine in methanol to concentrations of 1, 7.8125, 15.625, 31.25, 62.5, 125, 250, 500, and 1000 ng/mL. Subsequently, we analyzed these samples and plotted a

standard curve based on the relationship between peak area and concentration. This standard curve allowed us to accurately determine the concentration of 3-methylxanthine in the samples.

We have made corresponding supplements to the methods section (lines 787-794).

Question 3: The legend to figure 2B is confusing. Specifically, the colors indicate 'up' or 'down' but the notation indicates enriched in cisplatin or ABX+cisplatin mice. Please revise for consistency.

Response: We have addressed the issue you raised regarding the legend to Figure 2B. Specifically, we have revised the notation to ensure consistency. Now, the notations reads "Enriched in ABX+cisplatin" and "Enriched in cisplatin" to correspond with the colors indicating 'up' or 'down'. These changes should provide clarity and coherence to the figure.

Question 4: Figure 2C is still confusing to me. With the exception of the VIP score, this shows exactly the same information as Figure 2B. Why show both? This is unnecessarily confusing to readers of your manuscript.

Response: We appreciate your attention to our figures and their clarity. Our metabolite screening process indeed considers multiple factors, including fold change, p-value, and VIP score, to ensure a comprehensive analysis.

Figures 2B and 2C offer complementary perspectives on our results. Figure 2B emphasizes exceptional or significantly different values between the two groups,

focusing on differential fold change and p-values. On the other hand, Figure 2C provides a more specific representation by arranging metabolites based on their Log₂ fold change and highlighting their VIP scores. This approach offers a nuanced view of the differential metabolites.

Moreover, we recognize that Figure 2C, with its volcano plot format, uniquely presents the data, allowing for a clearer identification of significantly altered metabolites based on both fold change and statistical significance. We believe that the combination of these two figures effectively underscores the importance of 3-methylxanthine in our study.

Question 5: Figure 3A is confusing to me. The figure legend says that "the colors indicate the change of metabolites in the ABX+cisplatin group" but the legend within Figure 3A. Says that the colors indicate p-value. Which is it?

Response: Thank you for your review and valuable feedback. I apologize for any confusion caused by the unclear figure legend. Concerning Figure 3A, you are indeed correct in noting that the colors in the notation represent p-values. Regarding the legend of Figure 3B, which states "the colors indicate the change of metabolites in the ABX+cisplatin group," I understand your concern. The accurate interpretation is that the light red color represents the ABX+cisplatin group, while the light blue color represents the cisplatin group. To address this, we have also added appropriate labels to the notation in Figure 3B to reflect this clarification.

Question 6: Figure 3B is labeled as 'the caffeine metabolism pathway'. However, this figure only shows a select portion of the caffeine metabolism pathway in KEGG (https://www.genome.jp/kegg-bin/show_pathway?map00232). Please elaborate on what this is, and why only specific portions of the pathway have been shown.

Response: Thank you for your feedback. We regret any confusion caused by the labeling discrepancy in Figure 3B. The link provided directs to the comprehensive caffeine metabolism pathway in KEGG, while our figure presents only a partial perspective. Our rationale for this selective display is to spotlight metabolite abundances relevant to the caffeine metabolism pathway as detected in our metabolomics data, aligning with the primary focus of our study. Considering our emphasis on specific metabolite abundances, we have opted to highlight sections of the pathway where these metabolites are implicated. We have now updated the legend for Figure 3B(lines 1153-1154) to read: "Partial view of the caffeine metabolism pathway highlighting metabolite abundances detected in metabolomics data." Additionally, metabolites detected in our metabolomics data are annotated with a light green color for clarity. We appreciate your valuable feedback once again.

Question 7: Figure legend of figure 3F is not accurate. It says "The level of 3-methylxanthine in the feces of mice was quantified using HPLC." However, the figure is of an experimental design.

Response: Thank you for pointing out the discrepancy in the figure legend for Figure 3F. We apologize for the error. The legend incorrectly states that it describes the quantification of 3-methylxanthine levels in mouse feces using HPLC, while the figure

actually represents an experimental design. We have now revised the legend accordingly to accurately reflect the content:"Figure 3F: Experimental Design"(line 1158).Thank you for bringing this to our attention, and we appreciate your feedback.

Question 8: Please show individual values in all figures currently showed as column/bar charts (e.g., 1C, 1D, 1E, 3B, 3G-3I, 4B-4C, 4E, etc). Showing column plots with SEM suggest that the standard deviation is high.

Response: Thank you for your feedback. We have added individual values to all the column/bar charts as suggested, including figures such as 1C, 1D, 1E, 3B, 3G-3I, 4B, 4E, 5B-5D,6B-6G. This addition provides a more comprehensive view of the data distribution, beyond just the mean and standard error (SEM). Readers can now better assess the reliability and variability of the data with this improvement.

Re: mSystems01301-23R2 (**Gut-derived metabolite 3-methylxanthine enhances cisplatin-induced apoptosis via dopamine receptor D1 in a mouse model of ovarian cancer**)

Dear Mr. Xia Chen:

Your manuscript has been accepted, and I am forwarding it to the ASM production staff for publication. Your paper will first be checked to make sure all elements meet the technical requirements. ASM staff will contact you if anything needs to be revised before copyediting and production can begin. Otherwise, you will be notified when your proofs are ready to be viewed.

Cover Image Submissions: If you would like to submit a potential Cover Image, please email a file and a short legend to msystems@asmusa.org. Please note that we can only consider images that (i) the authors created or own and (ii) have not been previously published. By submitting, you agree that the image can be used under the same terms as the published article. Image File requirements: TIF/EPS, 7.5 inches wide by 8.25 inches tall (at least 2,250 pixels wide by 2,475 pixels tall), minimum 300 dpi resolution (600 dpi preferred), RGB, and no figure elements, e.g., arrows or panel labels. The legend should be a short description of the image, 1-2 sentences recommended.

Sincerely,
Neha Garg